# VS-Bench: Evaluating VLMs for Strategic Reasoning and Decision-Making in Multi-Agent Environments

## Abstract

Recent advancements in Vision Language Models (VLMs) have expanded their capabilities to interactive agent tasks, yet existing benchmarks remain limited to single-agent or text-only environments. In contrast, real-world scenarios often involve multiple agents interacting under rich visual and language observations, posing challenges with both multimodal perceptions and strategic interactions. To bridge this gap, we introduce Visual Strategic Bench (VS-Bench), a multimodal benchmark that evaluates VLM agents for strategic reasoning and decision-making in multi-agent environments. VS-Bench comprises eight vision-grounded environments spanning cooperative, competitive, and mixed-motive interactions, designed to assess agents' ability to infer other agents' future moves and optimize long-term objectives. We consider two complementary evaluation dimensions, including offline evaluation of strategic reasoning by next-action prediction accuracy and online evaluation of decision-making by normalized episode return. Extensive experiments of fourteen leading VLMs reveal a significant gap between current models and optimal performance, with the best model achieving 45.8% average prediction accuracy and 26.3% average normalized return. We further conduct in-depth analyses on multimodal input, social dilemma behaviors, and failure cases of VLM agents. By highlighting the limitations of existing models, we envision our work as a foundation for future explorations in strategic multimodal agents. Code and data are available at `https://sites.google.com/view/vs-bench-nips`.

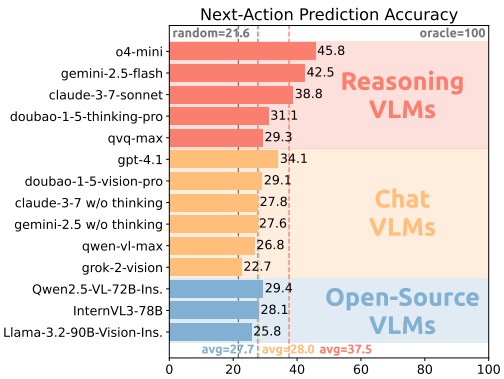 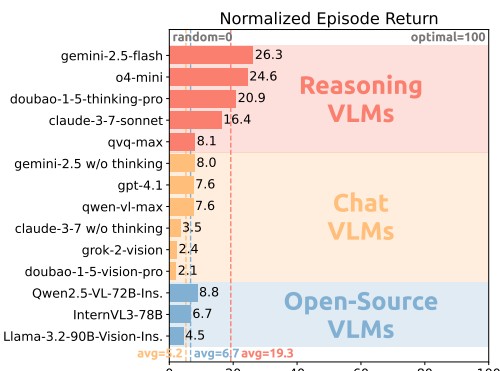

(a) Offline evaluation of strategic reasoning.      (b) Online evaluation of decision-making.

Figure 1: Evaluation results of fourteen VLMs' performance on strategic reasoning and decision-making averaged over eight environments in VS-Bench.

Submitted to 39th Conference on Neural Information Processing Systems (NeurIPS 2025). Do not distribute.

# 1 Introduction

Vision Language Models (VLMs) have recently unlocked impressive capabilities in open-world perception, multimodal reasoning, and interactive problem-solving [5, 39, 89]. Driven by these advancements, evaluations of VLMs have progressed beyond static tasks such as image captioning [15] and visual reasoning [3, 85] toward dynamic agent benchmarks including software engineering, computer use [30, 80], game environments [75, 87], and embodied control [25, 68, 83].

However, existing benchmarks for VLM agents mainly focus on single-agent settings, where one agent reasons and makes decisions in isolation. Yet the real world is inherently a multi-agent environment that involves cooperation, competition, and mixed-motive interactions, posing new challenges to the ability of intelligent agents [20, 77]. First, an agent's outcome depends not only on its own action but also on other agents' actions, requiring strategic reasoning to infer others' intentions and predict their future moves. Second, as all agents learn and adapt concurrently, the underlying dynamics become non-stationary, demanding agents to make decisions under uncertainty and optimize long-term objectives. Third, the coexistence of cooperation and competition gives rise to social dilemmas where agents must strategically balance self-interest and collective welfare. These challenges raise a crucial question that current benchmarks leave underexplored: ***How capable are VLM agents at strategic reasoning and decision-making in multi-agent environments?***

While prior efforts [1, 18, 79] have explored multi-agent evaluation for Large Language Models (LLMs), these benchmarks remain restricted to text-only environments, limiting their capability to assess agents in multimodal scenarios. On the one hand, many strategic domains such as board games [33, 64], card games [7, 11], and video games [8, 12] intrinsically rely on visual observations. Flattening these rich visual states into symbolic text strings requires hand-crafted encodings and inevitably discards spatial information critical for reasoning and decision-making. On the other hand, humans naturally integrate vision and language when interacting with others. Consequently, purely text-based environments diverge from real-world human-agent interactions and obscure progress toward developing human-compatible intelligent agents. These limitations underscore the need for a multimodal benchmark that incorporates visual context in multi-agent environments.

To bridge this gap, we introduce Visual Strategic Bench (VS-Bench), a multimodal benchmark designed to evaluate VLM for strategic reasoning and decision-making in multi-agent environments. VS-Bench comprises eight vision-grounded environments that cover three fundamental types of multi-agent interactions that emphasize different facets of strategic intelligence. (1) Cooperative games, including *Hanabi* and *Overcooked*, demand agents to understand teammates' intentions and coordinate their actions to achieve shared objectives. (2) Competitive games, including *Breakthrough*, *Kuhn Poker*, and *Atari Pong*, demand agents to model their opponents and stay robust against adversaries. (3) Mixed-motive games, including *Coin Dilemma*, *Monster Hunt*, and *Battle of the Colors*, demand agents to balance contradict interests and sustain cooperation while avoiding exploitation.

VS-Bench evaluates VLM agents along two complementary dimensions: offline evaluation of strategic reasoning and online evaluation of decision-making. **Strategic reasoning** refers to the theory-of-mind capability to infer other agents' intentions and predict their future moves for effective cooperation and competition. We construct an offline dataset for each environment and evaluate VLM agents' performance by their prediction accuracy of other agents' next actions. **Decision-making** focuses on agents' ability to optimize long-term objectives in non-stationary dynamics. We let VLM agents engage in online self-play or interactions with conventional agents in full-length episodes and evaluate their performance by normalized returns. By jointly analyzing both perspectives, our benchmark provides a unified and comprehensive evaluation of VLMs in multi-agent environments.

We evaluate fourteen leading VLMs, including three open-source models, six commercial chat models, and five commercial reasoning models on VS-Bench. Extensive results show that although current VLMs exhibit preliminary strategic reasoning ability by surpassing random agents, the best-performing model only attains a modest average prediction accuracy of $45.8\%$. Furthermore, current VLMs demonstrate poor decision-making ability in multi-agent environments, with the most capable model achieving a $26.3\%$ normalized return across all environments. Notably, although reasoning commercial models in general attain the best results, open-source models can achieve comparable performance to reasoning models in some mixed-motive games with prosocial behaviors for mutual benefit. We further conduct in-depth analyses to study the effect of multimodal input and prompting methods, the behaviors in social dilemmas, and the failure modes of VLM agents.

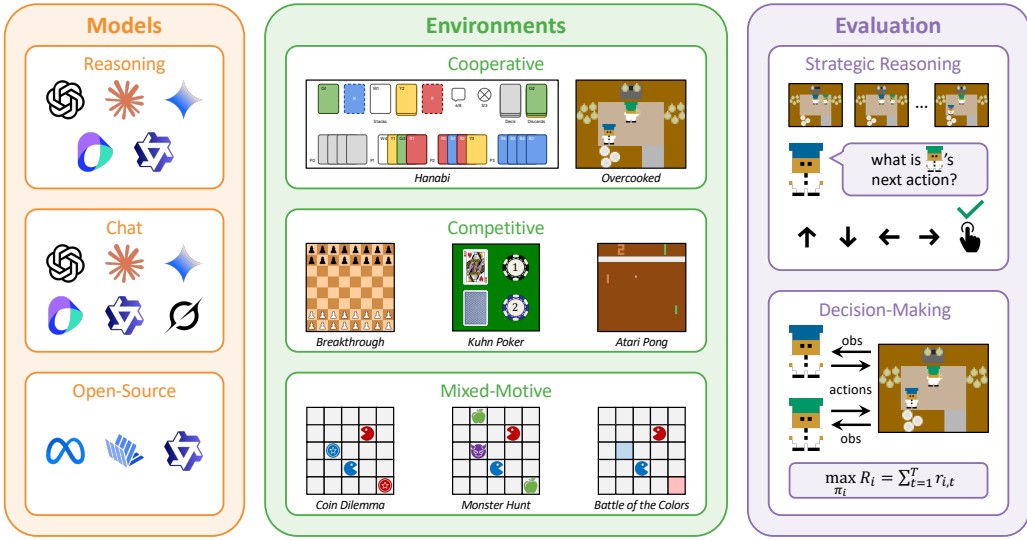

Figure 2: Overview of VS-Bench, a multimodal benchmark for evaluating VLMs in multi-agent environments. We evaluate fourteen state-of-the-art models in eight vision-grounded environments with two complementary dimensions, including offline evaluation of strategic reasoning by next-action prediction accuracy and online evaluation of decision-making by normalized episode return.

In summary, our contributions are threefold:

- We introduce VS-Bench, a multimodal benchmark for evaluating strategic reasoning and decision-making in multi-agent environments, comprising eight vision-grounded environments across cooperative, competitive, and mixed-motive interactions.

- We consider two complementary evaluation dimensions, including offline evaluation of strategic reasoning by next-action prediction accuracy and online evaluation of decision-making by normalized episode returns, to provide a unified and comprehensive assessment of VLM agents.

- We conduct extensive experiments of eleven commercial VLMs and three open-source VLMs and provide in-depth analyses of vision and language input, social behaviors, and failure modes, highlighting significant performance gaps for future research.

## 2 VS-Bench environments

In this section, we formalize the evaluation of VLMs in multi-agent environments and introduce eight vision-grounded games comprising VS-Bench. These games are carefully curated from classic game theory and multi-agent reinforcement learning (MARL), each serving as a well-recognized environment in the literature. We further adapt these games to incorporate image and text observations while preserving their strategic dynamics. By covering cooperative, competitive, and mixed-motive interactions, these games serve as a comprehensive benchmark for evaluating VLMs in multi-agent environments. A set of simpler games called VS-Bench Mini is described in Appendix A.

### 2.1 Problem formulation

Multi-agent environments are generally formulated as Partially Observable Markov Games (POMG) [38, 63]. A POMG is defined by a tuple $\mathcal{G} = (\mathcal{N}, \mathcal{S}, \{\mathcal{A}_i\}_{i \in \mathcal{N}}, \{\mathcal{O}_i\}_{i \in \mathcal{N}}, \mathcal{P}, \{\mathcal{R}_i\}_{i \in \mathcal{N}}, \gamma)$, where $\mathcal{N} = \{1, \cdots, n\}$ is the set of agents; $\mathcal{S}$ is the state space; $\mathcal{A}_i$ and $\mathcal{O}_i$ are the action space and observation space of agent $i$, respectively; $\mathcal{P} : \mathcal{S} \times \{\mathcal{A}_i\}_{i \in \mathcal{N}} \to \Delta(\mathcal{S})$ is the transition function; $\mathcal{R}_i : \mathcal{S} \times \{\mathcal{A}_i\}_{i \in \mathcal{N}} \to \mathbb{R}$ is the reward function of agent $i$; and $\gamma$ is the discount factor. In each step $t$, agent $i$ receives an observation $o_{i,t}$ and chooses an action $a_{i,t}$ according to its policy $\pi_i$. Given the current state $s_t$ and the joint action $a_t = (a_{1,t}, \cdots, a_{n,t})$, the environment transitions to the next state $s_{t+1} \sim \mathcal{P}(s_t, a_t)$ and each agent $i$ receive a reward $r_{i,t} = \mathcal{R}_i(s_t, a_t)$. The objective of agent $i$ is to maximize its expected accumulated reward $\mathbb{E}_{\pi_1, \cdots, \pi_n} [\sum_t \gamma^t r_{i,t}]$.

To evaluate VLM in multi-agent environments, we consider a multimodal observation space $\mathcal{O}_i = (\mathcal{I}_i, \mathcal{T}_i)$, where $\mathcal{I}_i$ is the space for image observations and $\mathcal{T}_i$ is the space for text prompts. We also consider a text-based action space $\tilde{\mathcal{A}}_i$ and a mapping function that converts each textual action into the original action space $\mathcal{A}_i$. To more comprehensively characterize the strategic ability of VLM agents, we consider three types of multi-agent interactions defined by the reward structure.

## 2.2 Cooperative games

In cooperative games, all agents share the same objective. Formally, the reward functions in cooperative games are identical: $\mathcal{R}_1(s, a) = \cdots = \mathcal{R}_n(s, a)$ for all $(s, a) \in \mathcal{S} \times \{\mathcal{A}_i\}_{i \in \mathcal{N}}$. To achieve strong performance in cooperative games, agents must understand their teammates' intentions under partial observability, divide the tasks to improve efficiency, and coordinate their actions to optimize the shared objective. We consider two representative cooperative games in MARL literature.

***Hanabi*** [7] is a partially-observable card game where players can observe others' cards but not their own. Each card has a color and a rank that can only be revealed through hint actions at the cost of an information token. To succeed, agents must coordinate to play cards in rank order for five colors. We consider the two-player full game, which is widely used for research on theory of mind, zero-shot coordination, and ad-hoc teamplay [27, 28]. Detailed descriptions can be found in Appendix B.1.

***Overcooked*** [23] is a popular video game where two chefs cooperate to cook and serve dishes in a kitchen. Each dish delivery requires multiple operations like navigating, chopping, cooking, and plating that are difficult to coordinate even for human players. Our implementation is based on Overcooked-AI [12], a well-known environment for zero-shot coordination and human-AI interactions [66, 86]. Detailed descriptions can be found in Appendix B.2.

## 2.3 Competitive games

In competitive games, the objective of each agent strictly contradicts with others. Formally, the reward functions in competitive games are zero-sum: $\sum_{i=1}^{n} \mathcal{R}_i(s, a) = 0$ for all $(s, a) \in \mathcal{S} \times \{\mathcal{A}_i\}_{i \in \mathcal{N}}$. To succeed in competitive games, agents must model their opponents to predict their future moves, stay robust against adversarial exploitation, and adapt to non-stationary dynamics. We consider three representative competitive games in game theory and MARL literature.

***Breakthrough*** [72] is a chess-like board game with simplified rules and identical pawns. Two players compete to advance their pieces across an $8 \times 8$ grid to reach the opponent's back row. The game is deceptively simple, yet it exhibits deep combinatorial complexity and sharp tempo imbalance between attack and defense, making it a suitable environment for studying multi-step lookahead and adversarial decision-making [41, 59]. Detailed descriptions can be found in Appendix B.3.

***Kuhn Poker*** [32] is a simplified variant of Texas Hold'em [48, 11] designed to study imperfect-information games for game-theoretic analysis. The game has a three-card deck and a single betting round where two players can either check or bet with limited stakes. Despite its minimal rules, Kuhn poker has been used as a classic game for counterfactual reasoning and decision-making with imperfect information [33, 49]. Detailed descriptions can be found in Appendix B.4.

***Atari Pong*** [4] is a classic arcade video game where two players control paddles to hit a ball across the screen. With raw pixel observations and competitive dynamics, Pong has become a canonical environment in the Arcade Learning Environment (ALE) [8] suite, which requires spatio-temporal reasoning and strategic gameplay [46, 47]. Detailed descriptions can be found in Appendix B.5.

## 2.4 Mixed-motive games

In mixed-motive games, agents' objectives are partially aligned and partially divergent. Formally, the reward functions are neither identical nor zero-sum, that is, there exists $(s, a)$ such that $\mathcal{R}_i(s, a) \neq \mathcal{R}_j(s, a)$ and $\sum_{i=1}^{n} \mathcal{R}_i(s, a) \neq 0$. To excel in mixed-motive games, agents must anticipate the hidden intentions of others, balance self-interest and common welfare, and achieve favorable equilibria. We consider three mixed-motive games adapted from classic social dilemmas in game theory.

***Coin Dilemma*** [35] is a grid-world environment inspired by the classic Prisoner's Dilemma [55] in game theory. A red player and a blue player move in a $5 \times 5$ grid world to collect red and blue coins. A player earns 1 point for collecting any coin. However, the blue player is penalized 2 points

if the red player collects a blue coin and vice versa. This setup creates a tension between mutual benefit and self-interest: while both players collecting their own color leads to a win-win result, unilateral defection maximizes one's own gains at the other's expense. Therefore, the game has been a common environment for studying rational reasoning, opponent shaping, and social dilemma resolution [21, 43, 58]. Detailed descriptions can be found in Appendix B.6.

*Monster Hunt* [53] is a grid-world environment inspired by the classic Stag Hunt [57] in game theory. Two players move in a $5 \times 5$ grid world to individually eat an apple for 2 points or jointly defeat a monster for 5 points. A player who confronts the monster alone, however, is penalized 2 points. This leads to multiple Nash equilibria where agents can both safely eat apples alone or take risks to cooperate for higher rewards. The game is used to investigate trust formation and risk-sensitive decision-making [34, 67]. Detailed descriptions can be found in Appendix B.7.

*Battle of the Colors* is a grid-world environment inspired by the classic Battle of the Sexes [44] in game theory. We propose and design this game in a manner similar to the previous two social dilemma games. A red player and a blue player move in a $5 \times 5$ grid world with a red block and a blue block. If both players move to the red block, the red player earns 2 points while the blue player earns 1 point, and vice versa. If players move to different color blocks, both players earn 0 points. Therefore, while coordination is mutually beneficial, each player strictly prefers coordinating on the block of their own color, creating a conflict of interest that produces two payoff-asymmetric Nash equilibria and a mixed equilibrium. This game thus challenges agents to solve conflicting preferences while avoiding coordination failure, making it suitable for studying equilibrium selection, bargaining dynamics, and social fairness. Detailed descriptions can be found in Appendix B.8.

## 3   Evaluating VLMs in multi-agent environments

To comprehensively benchmark VLMs in multi-agent environments, we consider two complementary dimensions including offline evaluation of strategic reasoning and online evaluation of decision-making. We further provide several insights from our evaluation, which highlight limitations of existing VLMs and research directions for future development.

**Model setup.** We select fourteen state-of-the-art VLMs for evaluation. For commercial VLMs, we select six chat models and five reasoning models from OpenAI GPT [50] and o-series [51], Anthropic Claude [2], Google Gemini [17], xAI Grok [78], Qwen [69], and Doubao [61]. For open-source VLMs, we select three leading models from Llama-3.2-Vision [45], InternVL3 [89], and Qwen2.5-VL [6]. We set the temperature to $1.0$ and the maximum number of output tokens to $8k$ for all models. We also set the maximum number of reasoning tokens to $16k$ for reasoning models. When encountering a cutoff for reaching maximum tokens, we dynamically extend the output and reasoning tokens to the model's limit. Detailed descriptions of model setups can be found in Appendix C.

### 3.1   Strategic reasoning

Strategic reasoning is the theory-of-mind ability to infer the hidden beliefs, desires, and intentions of other agents [31, 54]. This requires agents to think from others' perspectives and answer the question: *What would other agents do in the next steps?* Strategic reasoning is crucial in multi-agent environments because an agent's reward function depends not only on its own action, but also on others' actions. Therefore, to achieve strong performance, agents must anticipate teammates' moves to coordinate in cooperative games, predict opponents' actions to counter them in competitive games, and deduce whether to cooperate or compete in mixed-motive games. Detailed descriptions of strategic reasoning evaluation can be found in Appendix D.

**Evaluation setup.** We evaluate the strategic reasoning ability of VLM agents by their prediction accuracy of other agents' next actions on an offline dataset for each environment. More specifically, each sample in the dataset is a tuple $(\{\text{img}_{i,\tau}\}_{\tau=t'}^{t}, \text{text}_{i,t}, a_{-i,t+1})$, where $\{\text{img}_{i,\tau}\}_{\tau=t'}^{t}$ and $\text{text}_{i,t}$ are the image observation sequence and text prompt of agent $i$ at step $t$, respectively, and $a_{-i,t+1}$ is the action of other agents at the next step $t + 1$. The VLMs are prompted with the image sequence and text observations to predict the next actions of other agents. To ensure a rigorous and thorough evaluation, we construct the datasets according to the following principles. (1) Predictable: the next actions can be predicted from the image and text observations, which exclude actions like reveal in *Hanabi* that requires unobservable information. (2) Diverse: the dataset should cover a diverse

| Models[1] | Overall | Cooperative | | Board [2] | Comptitive | | Mixed-Motive | | |
|---|---|---|---|---|---|---|---|---|---|
| | | *Hanabi* | *Overcooked* | | *Poker* | *Pong* | *Dilemma* | *Hunt* | *Battle* |
| Oracle | 100.0 | 100.0 | 100.0 | 100.0 | 100.0 | 100.0 | 100.0 | 100.0 | 100.0 |
| o4-mini | 45.8 | 58.3 | 31.8 | 26.8 | 63.5 | 43.5 | 53.5 | 36.6 | 52.5 |
| gemini-2.5-flash | 42.5 | 37.0 | 21.0 | 23.3 | 65.0 | 41.3 | 57.5 | 31.2 | 63.5 |
| claude-3-7-sonnet | 38.8 | 39.0 | 26.0 | 24.3 | 65.5 | 44.8 | 45.0 | 26.2 | 39.5 |
| doubao-1-5-thinking-pro | 31.1 | 32.8 | 26.3 | 19.8 | 57.8 | 44.3 | 22.8 | 18.4 | 27.0 |
| qvq-max | 29.3 | 32.3 | 19.0 | 21.8 | 59.3 | 37.8 | 24.3 | 16.8 | 23.5 |
| gpt-4.1 | 34.1 | 23.0 | 27.0 | 22.5 | 54.0 | 41.5 | 40.3 | 30.0 | 34.8 |
| doubao-1-5-vision-pro | 29.1 | 15.0 | 22.3 | 15.8 | 53.8 | 31.3 | 33.8 | 32.3 | 28.8 |
| claude-3-7 w/o thinking | 27.8 | 9.8 | 16.0 | 18.0 | 56.0 | 43.3 | 26.8 | 25.8 | 26.8 |
| gemini-2.5 w/o thinking | 27.6 | 21.5 | 19.3 | 14.8 | 48.5 | 34.0 | 32.0 | 23.0 | 27.5 |
| qwen-vl-max | 26.8 | 26.5 | 26.0 | 19.5 | 45.3 | 23.5 | 25.8 | 23.5 | 24.3 |
| grok-2-vision | 22.7 | 12.8 | 17.3 | 10.8 | 53.3 | 20.8 | 24.5 | 22.0 | 20.5 |
| Qwen2.5-VL-72B-Ins. | 29.4 | 26.8 | 26.5 | 23.8 | 45.2 | 27.0 | 28.8 | 27.2 | 30.0 |
| InternVL3-78B | 28.1 | 25.3 | 20.5 | 14.0 | 45.5 | 34.8 | 35.8 | 23.3 | 25.5 |
| Llama-3.2-90B-Vision-Ins. | 25.8 | 20.0 | 16.5 | 11.8 | 53.3 | 36.3 | 25.8 | 24.0 | 18.8 |
| Random | 21.6 | 8.8 | 16.7 | 4.3 | 50.0 | 33.3 | 20.0 | 20.0 | 20.0 |

Table 1: Strategic reasoning evaluation results. For each environment, the first , second , and third best results are highlighted in green, while the results below random are highlighted in red.

range of observations and actions in the environments, considering different environment contexts and different styles of other agents. (3) Balanced: the distribution of the samples should be balanced to avoid bias toward certain behaviors or preferences. Guided by these principles, we collect a dataset of 400 samples for each environment and benchmark fourteen VLMs for strategic reasoning ability measured by next-action prediction accuracy in eight environments.

The evaluation results in Table 1 and Fig. 1a show that current VLMs exhibit certain strategic reasoning ability by surpassing random in overall prediction accuracy, yet they still lag behind the oracle results by a noticeable margin of about 50%. All fourteen models perform better than random guessing in at least six of the eight games, demonstrating non-trivial theory-of-mind capability in multi-agent environments. Reasoning models generally achieve better results than chat models and open-source models, with the best-performing model o4-mini attaining an overall accuracy of 45.8% and consistently ranking in the top three across all environments. Notably, the three leading open-source models achieve an average overall accuracy of 27.7%, which is comparable to the commercial chat models with a 28.0% average overall accuracy. However, even these most capable existing models attain less than 50% overall accuracy, leaving a 50% gap to the oracle. This deficit is especially pronounced in *Overcooked*, *Leduc Poker*, *Atari Pong*, and *Monster Hunt*, three of which are adapted from video games. We further investigate this observation in the next analysis section.

> **Finding 1:** Existing VLMs exhibit preliminary strategic reasoning ability by outperforming random guessing in most environments, yet the 50% gap between the most capable models and oracle results remains to be narrowed for future research.

## 3.2 Decision-making

Decision-making is the ability to optimize one's long-term objectives under uncertainty [19]. This requires agents to prioritize future accumulated returns over immediate gains, adapt to non-stationary dynamics with evolving agents, and balance cooperation and competition to navigate toward favorable equilibria. Detailed descriptions of decision-making evaluation can be found in Appendix E.

**Evaluation setup.** We evaluate the decision-making ability of VLM agents by their normalized episode returns through online self-play or interactions with conventional agents in each environment. More specifically, for cooperative and mixed-motive games, we let multiple VLM agents of the same type interact with each other. For competitive games, we evaluate VLM agents against well-recognized conventional agents like Monte Carlo Tree Search (MCTS) [16], Counterfactual Regret Minimization (CFR) [90], and Atari built-in bot [8]. For all environments, we also evaluate the

---

[1]Specific model versions and links to open-source models can be found in Appendix C.

[2]The *Board* column corresponds to *Breakthrough*, and the remaining columns to its right correspond to *Kuhn Poker*, *Atari Pong*, *Coin Dilemma*, *Monster Hunt*, and *Battle of the Colors*, respectively.

| Models | Overall | Cooperative | | Board | Comptitive | | Mixed-Motive | | |
| | | *Hanabi* | *Overcooked* | *Board* | *Poker* | *Pong* | *Dilemma* | *Hunt* | *Battle* |
|---|---|---|---|---|---|---|---|---|---|
| Optimal | 100.0 | 100.0 | 100.0 | 100.0 | 100.0 | 100.0 | 100.0 | 100.0 | 100.0 |
| gemini-2.5-flash | 26.3 | $27.1_{\pm36.0}$ | $8.5_{\pm5.4}$ | $20.0_{\pm51.5}$ | $84.1_{\pm19.9}$ | $1.6_{\pm1.9}$ | $10.0_{\pm25.5}$ | $26.2_{\pm5.8}$ | $32.8_{\pm8.5}$ |
| o4-mini | 24.6 | $42.9_{\pm30.5}$ | $17.0_{\pm6.8}$ | $30.0_{\pm94.0}$ | $71.6_{\pm21.1}$ | $11.2_{\pm16.5}$ | $-4.6_{\pm21.4}$ | $24.9_{\pm8.2}$ | $3.5_{\pm5.4}$ |
| doubao-1-5-thinking-pro | 20.9 | $56.7_{\pm22.8}$ | $10.1_{\pm4.7}$ | $10.0_{\pm42.0}$ | $65.8_{\pm4.9}$ | $2.9_{\pm2.5}$ | $0.7_{\pm3.2}$ | $17.2_{\pm11.3}$ | $4.0_{\pm4.8}$ |
| claude-3-7-sonnet | 16.4 | $6.7_{\pm21.1}$ | $10.1_{\pm3.5}$ | $20.0_{\pm79.5}$ | $67.7_{\pm28.1}$ | $-0.5_{\pm1.0}$ | $4.6_{\pm15.4}$ | $19.9_{\pm3.5}$ | $2.5_{\pm4.6}$ |
| qvq-max | 8.1 | $0.0_{\pm0.0}$ | $2.0_{\pm3.4}$ | $5.0_{\pm31.5}$ | $57.2_{\pm19.4}$ | $0.4_{\pm1.6}$ | $0.0_{\pm2.1}$ | $0.7_{\pm4.5}$ | $-0.5_{\pm0.0}$ |
| gemini-2.5 w/o thinking | 8.0 | $0.0_{\pm0.0}$ | $2.0_{\pm4.0}$ | $0.0_{\pm0.0}$ | $58.6_{\pm12.2}$ | $1.0_{\pm1.4}$ | $-0.7_{\pm4.3}$ | $0.7_{\pm8.9}$ | $2.5_{\pm3.4}$ |
| gpt-4.1 | 7.6 | $0.0_{\pm0.0}$ | $-0.5_{\pm0.0}$ | $0.0_{\pm0.0}$ | $31.9_{\pm10.2}$ | $0.2_{\pm1.4}$ | $17.8_{\pm6.7}$ | $11.2_{\pm5.6}$ | $0.5_{\pm2.0}$ |
| qwen-vl-max | 7.6 | $1.2_{\pm2.0}$ | $-0.5_{\pm0.0}$ | $0.0_{\pm0.0}$ | $47.6_{\pm8.6}$ | $-0.3_{\pm1.0}$ | $-0.4_{\pm2.8}$ | $13.2_{\pm20.2}$ | $-0.5_{\pm0.0}$ |
| claude-3-7 w/o thinking | 3.5 | $0.0_{\pm0.0}$ | $2.0_{\pm4.0}$ | $5.0_{\pm31.5}$ | $19.1_{\pm17.8}$ | $-0.9_{\pm0.3}$ | $1.4_{\pm9.2}$ | $0.2_{\pm8.2}$ | $1.0_{\pm2.3}$ |
| grok-2-vision | 2.4 | $0.0_{\pm0.0}$ | $1.5_{\pm3.4}$ | $0.0_{\pm0.0}$ | $16.6_{\pm8.1}$ | $-0.1_{\pm1.5}$ | $1.1_{\pm7.0}$ | $-0.4_{\pm5.8}$ | $0.5_{\pm2.0}$ |
| doubao-1-5-vision-pro | 2.1 | $0.0_{\pm0.0}$ | $-0.5_{\pm0.0}$ | $0.0_{\pm0.0}$ | $13.4_{\pm28.7}$ | $-0.9_{\pm0.3}$ | $-2.1_{\pm5.2}$ | $7.8_{\pm8.2}$ | $-0.5_{\pm0.0}$ |
| Qwen2.5-VL-72B-Ins. | 8.8 | $0.8_{\pm1.8}$ | $-0.5_{\pm0.0}$ | $0.0_{\pm0.0}$ | $52.0_{\pm13.1}$ | $-0.8_{\pm0.2}$ | $0.0_{\pm2.7}$ | $19.6_{\pm25.7}$ | $-0.5_{\pm0.0}$ |
| InternVL3-78B | 6.7 | $0.0_{\pm0.0}$ | $0.0_{\pm1.5}$ | $0.0_{\pm0.0}$ | $49.8_{\pm17.5}$ | $-0.9_{\pm0.3}$ | $6.8_{\pm8.9}$ | $-1.8_{\pm9.2}$ | $0.0_{\pm1.5}$ |
| Llama-3.2-90B-Vision-Ins. | 4.5 | $0.0_{\pm0.0}$ | $1.5_{\pm3.3}$ | $0.0_{\pm0.0}$ | $30.1_{\pm8.7}$ | $-0.9_{\pm0.3}$ | $0.4_{\pm3.4}$ | $3.6_{\pm4.9}$ | $1.0_{\pm2.3}$ |
| Random | 0.0 | 0.0 | 0.0 | 0.0 | 0.0 | 0.0 | 0.0 | 0.0 | 0.0 |

Table 2: Decision-making evaluation results. For each environment, the first , second , and third best results are highlighted in green, while the results below or equal to random are in red.

random agents and the oracle agents with maximum return to normalize the results so that the normalized return for random agents is 0 and the normalized return for oracle agents is 100.

The evaluation results in Table 2 and Fig. 1b show that current VLMs are poor at decision-making in multi-agent games, with a significant gap of about 75% normalized return behind optimal agents. As illustrated by the large swaths of red cells, twelve out of fourteen evaluated models perform no better than random agents in at least one environment, indicating their incompetence to optimize long-term return in the face of non-stationary, interdependent multi-agent dynamics. Although reasoning models achieve relatively better results than chat models and open-source models, even the most capable model gemini-2.5-flash only attains an overall normalized return of 26.3%, which is far behind the optimal normalized return. Surprisingly, we observe that some open-source models can achieve comparable results to reasoning models in certain mixed-motive games like Qwen2.5-VL-72B-Ins. in *Coin Dilemma* and InternVL3-78B in *Monster Hunt*. We also observe that the cases where models fall below random performance are concentrated on video games like *Overcooked*, *Atari Pong*, and *Coin Dilemma*, which underscores the coupled difficulty of multimodal perception and strategic decision-making. We further investigate and analyze these observations in the next section.

> **Finding 2:** Existing VLMs exhibit poor decision-making ability in multi-agent environments, highlighting a significant gap of 75% that remains an open challenge for future research.

## 4 Analysis

Motivated by the observations in the evaluation results, we further investigate several aspects of VLMs in multi-agent environments and provide in-depth analyses on multimodal input, social behaviors, and failure cases. More experiment results can be found in Appendix F.

### 4.1 Multimodal input

In principle, multimodal observations provide more information and should lead to better strategic reasoning and decision-making. However, we observe in the evaluation results that environments with inherent visual states, like video games, are especially challenging for VLM agents, indicating potential incompetence in multimodal environments. To investigate, we select three games–a card game, a board game, and a video game–and perform ablations on both vision and language input.

We first replace image inputs with text descriptions and compare the strategic reasoning results. The first row in Fig. 3 shows that, with image input, reasoning models' average performance increases in the video game but decreases in the other two games. This indicates that VLMs can fail to utilize multimodal inputs for better performance. Next, we consider Chain-of-Thought (CoT) prompting [76] and the second row in Fig 3 shows CoT improves chat models' performance in card and board games

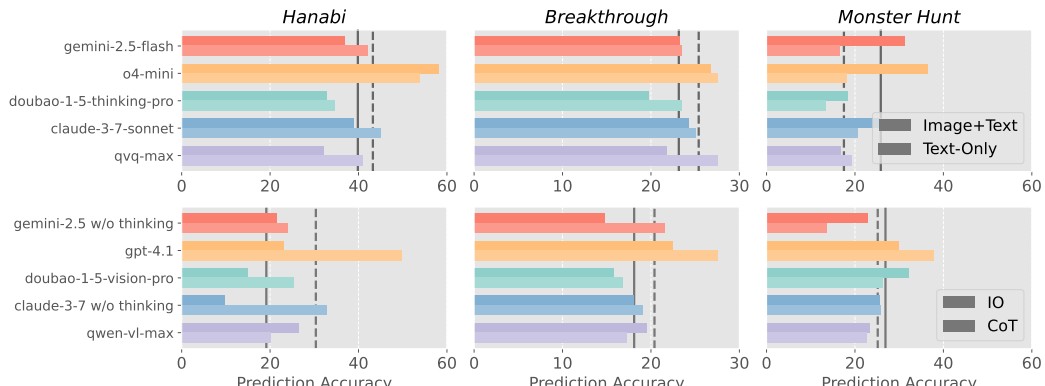

Figure 3: Ablations on visual input (first row) and prompting method (second row). Vertical solid and dashed lines represent average results for the default and ablation settings, respectively.

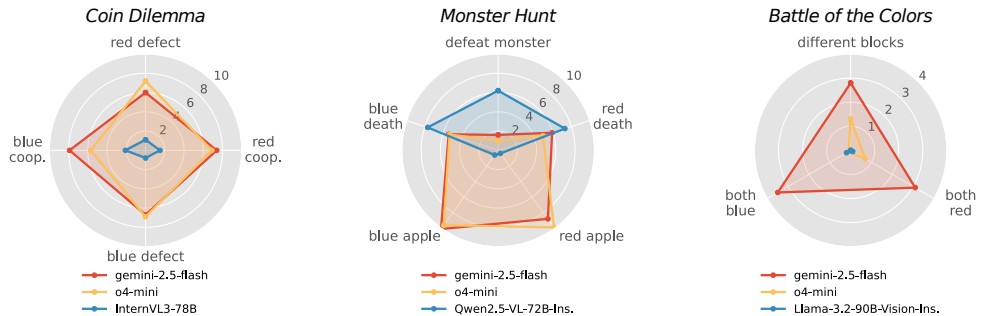

Figure 4: Behaviors of two reasoning models and the best-performing open-source models in mixed-motive social dilemma games. Dimensions are agents' behaviors described in Sec. 2.4.

but not in video games, showing VLMs' failure to perform step-by-step reasoning in visual-rich environments for better performance. More evaluation results can be found in Appendix F.1.

> **Finding 3:** Existing VLMs can fail to improve performance with visual observations and CoT prompting, underscoring their incompetence in multimodal environments.

## 4.2 Behaviors in social dilemma

Another interesting observation is that open-source models can achieve comparable results to reasoning models in some mixed-motive games. We investigate this by visualizing the behaviors of two reasoning models and the best-performing open-source models in each social dilemma games. As shown in Fig 4, in *Coin Dilemma*, the reasoning models are better at collecting coins, as they cooperate (collect their own coin) and defect (collect others' coin) more times than the open-source model. However, they are also more self-interested, especially `o4-mini`, which tends to collect others' coins instead of its own, resulting in a worse-than-random result. In comparison, although `InterVL3-78B` is not adept at collecting coins, it exhibits a strong preference for collecting its own coins rather than those of others, leading to a win-win situation where both agents get high returns. Similar behaviors can be found in *Monster Hunt*, where reasoning models like `gemini-2.5-flash` tend to safely eat apples alone and avoid encountering the monster. By contrast, `Qwen2.5-VL-72B-Ins.` is more inclined to take the risk to cooperate and defeat the monster together, which gives a high reward. More results on social behavior analysis can be found in Appendix F.2.

> **Finding 4:** Open-source VLMs can achieve comparable results to commercial reasoning VLMs in some social dilemma games with prosocial behaviors for mutual benefit.

### 4.3 Failure case analysis

To understand why VLMs underperform in multi-agent environments, we conduct a qualitative analysis of their failure cases. In strategic reasoning, two common failure cases are ignoring history and private information. For example, in *Hanabi*, players' cards are observable to other agents but not to themselves. VLMs often overlook this information asymmetry and incorrectly use their private information to predict the next actions of others. In decision-making, another common failure case is focusing excessively on one's own actions while ignoring those of others. For example, in *Breakthrough*, VLMs tend to persistently advance their own pieces and fail to identify defensive vulnerabilities that directly result in losing the match. More failure cases can be found in Appendix G.

## 5 Related work

### 5.1 Multi-agent environments and benchmarks

Early work on multi-agent reasoning and decision-making is grounded in game theory [22, 74], which models interactions among rational players and introduces canonical testbeds like board games [62, 70], card games [32, 65], and social dilemmas [44, 55, 57]. Building on these foundations, breakthroughs in multi-agent reinforcement learning (MARL) [11, 64] have expanded the field toward complex, high-dimensional environments covering a diverse range of cooperative [7, 12, 60], competitive [48, 73], and mixed-motive tasks [42, 9]. Despite their impressive achievements, agents developed in these environments are typically specialized for a single task and lack general-purpose abilities to perform strategic reasoning and decision-making across different domains.

Recent advancements in Large Language Models (LLMs) [24, 52, 71] have catalyzed a paradigm shift toward generalist agents that can perceive and act in various environments without task-specific training. A growing body of text-based benchmarks has been proposed to evaluate different facets of LLM agents in multi-agent environments covering cooperation [1], competition [18, 29], and mixed-motive interactions [14, 79, 81]. However, these benchmarks mainly focus on text-only environments, which do not align with real-world decision-making that integrates visual observation, spatial reasoning, and multimodal context. Our work fills this gap by introducing eight vision-grounded games to evaluate multimodal generalist agents in multi-agent environments.

### 5.2 VLM agent benchmarks

The rapid evolution of Vision Language Models (VLMs) [5, 39] has driven evaluation beyond static tasks like image captioning [15] and visual reasoning [3, 85] toward interactive agent environments. Existing benchmarks can be broadly categorized into four domains: coding, GUI interaction, game environments, and embodied control. Coding benchmarks [13, 36, 82] consider software engineering and machine learning engineering with both visual and text input. GUI benchmarks evaluate VLMs on graphic interface operations like web browsing [26, 30, 88], computer use [80, 10], and phone use [37, 56]. Game benchmarks [40, 75, 87] offer dynamic virtual environments with structured rewards to assess VLMs' ability in perception, reasoning, and decision-making. Embodied benchmarks [25, 68, 83] evaluate VLMs in vision-driven robotics control and physical world interactions. Nevertheless, these benchmarks predominantly concentrate on single-agent tasks, which overlook the distinctive challenges of multi-agent environments including non-stationary dynamics, interdependent decision-making, and equilibrium selection. Our work bridges this gap by evaluating VLMs in multi-agent games with both offline evaluation of strategic reasoning and online evaluation of decision-making.

## 6 Conclusion

In this work, we present VS-Bench, a comprehensive multimodal benchmark for evaluating strategic reasoning and decision-making capabilities of VLMs in multi-agent environments. Through eight vision-grounded environments and two complementary evaluation metrics of next-action prediction accuracy and normalized episode returns, we establish a unified framework for assessing VLMs in multi-agent interactions. Extensive experiments and analysis on fourteen state-of-the-art VLMs reveal a significant gap between current models and optimal performance, highlighting their limitations for future development. By releasing VS-Bench as an open platform, we aim to spur research on strategic multimodal agents that excel in vision-grounded multi-agent environments.

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

## A  VS-Bench Mini

We introduce a lightweight benchmark, VS-Bench Mini, for the preliminary evaluation of VLMs. The benchmark comprises one simple cooperative game, *Tiny-Hanabi*, and one competitive game, *Tic-Tac-Toe*. Figure 5 presents example visual inputs for these two games.

Standard two-player *Hanabi* is played with a hand size of 5 cards per player, five colors, and a maximum rank of 5, drawn from a 50-card deck. For *Tiny-Hanabi*, we simplify the configuration to a hand size of 3 cards and a maximum rank of 3 per color, using only two colors. Both configurations employ three life tokens (penalties for misplays) and eight information tokens (used to convey hints). Under the *Tiny-Hanabi* setting, VLMs can focus more on cooperative strategy, with reduced complexity in inferring cards and colors.

*Tic-Tac-Toe* is a fundamental competitive board game played on a $3 \times 3$ grid, in which the first player to align three of their marks horizontally, vertically, or diagonally wins. We implement an optimal Monte Carlo Tree Search (MCTS) agent configured with an exploration constant $c = 2.0$, a maximum of 1000 simulations per move, and 10 rollouts per move. However, since *Tic-Tac-Toe* yields only drawn outcomes under optimal play, superior models are characterized by their ability to achieve a higher proportion of draws when matched against our MCTS agent.

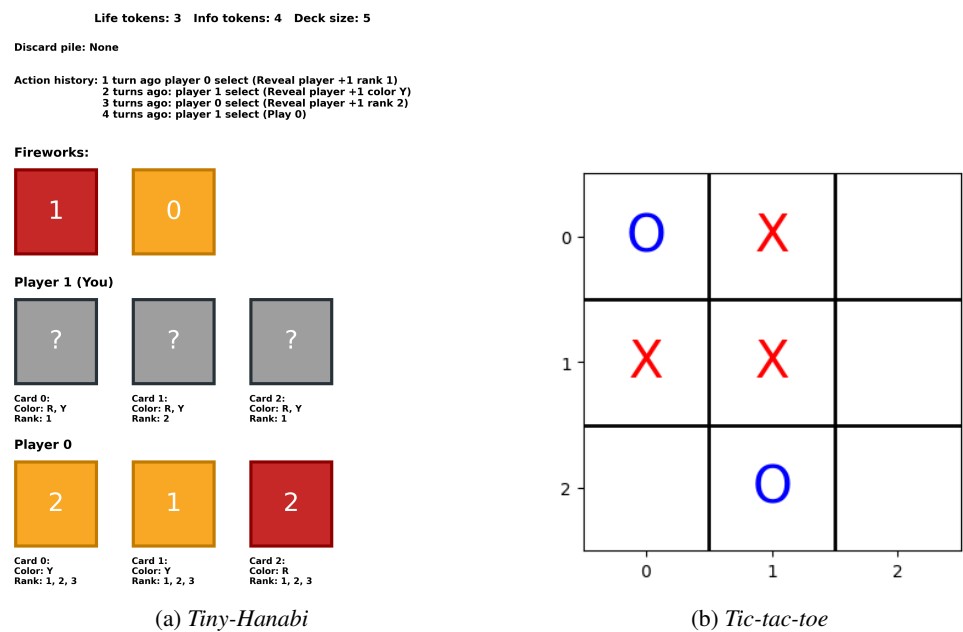

(a) *Tiny-Hanabi*                    (b) *Tic-tac-toe*

Figure 5: Example visual inputs of the two games in VS-Bench Mini.

## B  Environment details

### B.1  Hanabi

**Visual observation.** An example is shown in Figure 6. The current game-state visualization is divided into four principal sections:

1. **Basic Information**: counts of remaining life tokens, information tokens, and draw deck size;
2. **Historical Information**: all discarded cards, together with each player's two most recent actions;
3. **Fireworks**: current progress of each color's fireworks stack, indicating the highest played rank;
4. **Players' Hands**: each player's hand, with annotations beneath each card showing all possible colors and ranks deduced from received hints.

Figure 6: Example visual input of *Hanabi*.

Figure 7: Example visual input of *Overcooked*.

**System prompt.**

```
You are an AI agent that makes optimal decisions in the game of hanabi.
```

**User prompt.**

```
GAME RULES:
1. Hanabi is a cooperative card game for 2 players.
2. The deck consists of 5 colors: R(Red), Y(Yellow), G(Green), W(White),
↪  B(Blue), with ranks ranging from 1 to 5. Each color contains 10 cards: three
↪  of rank 1, two each of rank 2 through 4, and one of rank 5, for a total of
↪  50 cards.
3. Each player holds 5 cards in hand.
4. There are 8 Info tokens (used to give hints) and 3 Life tokens (penalties for
↪  misplays).
5. As in blind man's bluff, players can see each other's cards but they cannot
↪  see their own. Play proceeds around the table; each turn, a player must take
↪  one of the following actions:
   a. (Play i): play the i-th card from your hand (0-indexed) and attempt to
   ↪  add it to the cards already played. This is successful if the card is a
   ↪  1 in a suit that has not yet been played, or if it is the next number
   ↪  sequentially in a suit that has been played. Otherwise a Life token is
   ↪  consumed and the misplayed card is discarded. Successfully playing a 5
   ↪  of any suit replenishes one Info token. Whether the play was successful
   ↪  or not, the player draws a replacement card from the deck (if any
   ↪  remain).
   b. (Discard i): discard the i-th card from your hand and draw a replacement
   ↪  card from the deck (if any remain). The discarded card is out of the
   ↪  game and can no longer be played. Discarding a card replenishes one Info
   ↪  token.
   c. (Reveal player +1 color c): spend one Info token to reveal all cards of
   ↪  color c in the other player's hand.
   d. (Reveal player +1 rank r): spend one Info token to reveal all cards of
   ↪  rank r in the other player's hand.
6. The game ends immediately when either all Life tokens are used up, resulting
↪  in a game loss with a score of 0, or when all 5s have been successfully
↪  played, resulting in a game win with a score of 25. Otherwise, the game
↪  continues until the deck runs out and one final round is completed. At the
↪  end of the game, the final score is calculated as the sum of the highest
↪  card played in each suit, up to a maximum of 25 points.
```

**User prompt continued.**

```
PLAYER INFORMATION:
You are player {Player ID}.

GAME STATE:
Below is a visual representation of the current game state:
    - The first section, located above the image, presents the game's basic
    ↪  state information.
    - The second section summarizes the most recent player actions.
    - The third section displays the current firework stacks, with each color
    ↪  labeled by the highest successfully played rank.
    - The fourth section shows your own hand, represented as gray squares marked
    ↪  with '?', reflecting the fact that you cannot see your own cards.
    - The fifth section presents the other player's hand, with each card shown
    ↪  in its true color and rank, since it is fully visible to you.
Below each card, you will find two lines of inferred information:
    - Color: a list of all possible colors deduced for that card so far.
    - Rank: a list of all possible ranks deduced for that card so far.
The information displayed below your cards reflects the hints the other player
↪  has given you so far.
The information below the other player's cards represents what they currently
↪  believe about their own cards, based on all the useful hints you have
↪  provided them up to this point. For example, below your first card you might
↪  see:
    Card 0:
    Color: R, Y
    Rank: 2, 3
indicating that your card 0 is either Red or Yellow and has rank 2 or 3.

LEGAL ACTIONS:
{Current Legal Actions}

INSTRUCTIONS:
Now it is your turn to choose an action. You should output your action in the
↪  following JSON format:
```json
{
    "action": "(ACTION)"
}
```
where (ACTION) is one of the actions listed in the LEGAL ACTIONS section.

Do not include any extra commentary or explanation.
```

Back to cooperative games.

## B.2  Overcooked

**Visual observation.** An example is shown in Figure 7. On the left is the current game state, showing the overall kitchen layout, the positions and orientations of both chefs, and the items they hold. On the right is a legend explaining the visual representations of game elements—such as objects and chef orientations—used in the game state.

**System prompt.**

```
You are an AI agent that makes optimal decisions in the game of Overcooked.
```

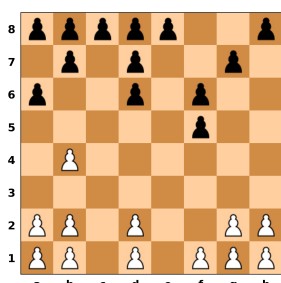
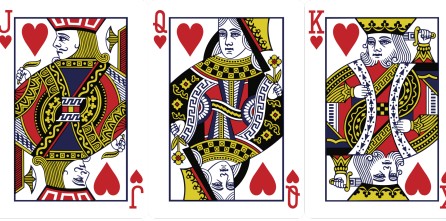
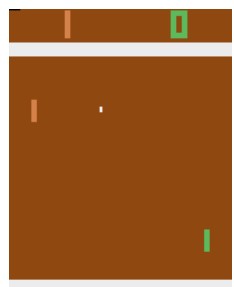

(a) Example visual input of *Breakthrough*.

(b) Example visual input of *Kuhn Poker*.

(c) Example visual input of *Atari Pong*.

## User prompt.

```
GAME RULES:
1. Overcooked is a cooperative game where two chefs collaborate to cook and
↪  serve soups in 50 timesteps.
2. The chefs can move in the available area and cannot move to the counter.
3. The chefs can interact with the object on the tile that they are facing.
4. A soup is cooked in the following steps:
    a. Pick up (interact) 1 onion and place (interact) it in the pot.
    b. After placing 3 onions in the pot, open (interact) the pot and cook for 5
    ↪  timesteps. The pot will show how long the soup has been cooked.
    c. When the pot shows the number 5, the soup is finished. Pick up (interact)
    ↪  a dish to plate (interact) the soup.
    d. Deliver the soup and put (interact) it on the serving location.

PLAYER INFORMATION:
1. You are controlling {Chef ID} in the {Hat Color}.
2. You are holding {Holding Text} currently.
3. The image sequence shows the 4 most recent game frames, with the last image
↪  being the current game frame. Each image shows the frame and object legend,
↪  with the timestep in the top left corner.

HISTORY ACTIONS:
{History Informations}
(e.g., In timestep 1: chef_0 chooses {Action}, chef_1 chooses {Action})

LEGAL ACTIONS:
1. <UP>: face up and move up one tile if possible.
2. <DOWN>: face down and move down one tile if possible.
3. <RIGHT>: face right and move right one tile if possible.
4. <LEFT>: face left and move left one tile if possible.
5. <STAY>: stay in the current tile and do nothing.
6. <INTERACT>: interact with the object on the tile that you are facing.

INSTRUCTIONS:
Now you should choose an action base on the game state in the current game
↪  frame. You should output your action in the following JSON format:
```json
{
    "action": "<ACTION>"
}
```
where <ACTION> is one of <UP>, <DOWN>, <LEFT>, <RIGHT>, <STAY>, <INTERACT>.

Do not include any extra commentary or explanation.
```

Back to cooperative games.

## B.3 Breakthrough

**Visual observation.** An example is shown in Figure 8a. The figure illustrates the current positions of both black and white pieces on the board. Row and column indices are annotated on the left and bottom sides of the image, respectively.

**System prompt.**

```
You are an AI agent that makes optimal decisions in the game of breakthrough.
```

**User prompt.**

```
GAME RULES:
1. Breakthrough is a two-player strategy game played on a 8x8 grid.
2. Each player controls pieces of a color: 'White' or 'Black'. 'White' starts at
↪   the bottom (rows 1 and 2), while 'Black' starts at the top (rows 7 and 8).
3. If 'White' moves a piece to row 8, 'White' wins the game. Conversely, if
↪   'Black' moves a piece to row 1, 'Black' wins the game.
4. Players alternate turns, moving one piece per turn, with 'Black' going first.
5. A piece may only move one space straight or diagonally forward, and only if
↪   the destination square is empty.
6. A piece may only capture an opponent's piece by moving one space diagonally
↪   forward into its square. In this case, the opponent's piece is removed, and
↪   your piece takes its place.
7. 'Black' moves forward by decreasing row indices (downward), while 'White'
↪   moves forward by increasing them (upward).
8. Moves are specified by their start and end positions. For example, 'a2a3'
↪   indicates moving a piece from a2 (column a, row 2) to a3 (column a, row 3).
9. The board is labeled with columns a-h and rows 1-8. Thus, h8 is the top-right
↪   corner, and a1 is the bottom-left corner.

PLAYER INFORMATION:
Your mark is {Player's Mark}.

GAME STATE:
The current grid is shown in the image. Row labels are displayed on the left,
↪   while column labels appear at the bottom. The pieces are marked using their
↪   corresponding colors in the grid.

LEGAL ACTIONS:
{Legal Actions}

INSTRUCTIONS:
It is now your turn to select an action. Please output your move in the
↪   following JSON format:
```json
{
    "action": "xiyj"
}
```
where:
- "x" and "y" represent the column letters, ranging from 'a' to 'h'.
- "i" and "j" represent the row numbers, ranging from 1 to 8.

For example, "a2a3" means moving the piece from column 'a', row 2 to column 'a',
↪   row 3.

Do not include any extra commentary or explanation.
```

Back to competitive games.

## B.4  Kuhn Poker

**Visual observation.** An example is shown in Figure 8b. Each player receives a visual representation of their actual card based on the true information of their hand.

**System prompt.**

```
You are an AI agent that makes optimal decisions in the game of Kuhn poker.
```

**User prompt.**

```
GAME RULES:
1. Kuhn poker is a two-player card game. The deck includes only three cards:
↪  King (K) > Queen (Q) > Jack (J).
2. At the start of each game, both player 0 and player 1 place 1 chip into the
↪  pot as a blind ante.
3. Each player is dealt a card as private information, and the third card is set
↪  aside unseen.
4. The two players take turns acting, starting with player 0. A player can
↪  choose to:
     a. <PASS>: place no additional chips into the pot.
     b. <BET>: place 1 additional chip into the pot.
5. If a player chooses to <PASS> after the other player's <BET>, the betting
↪  player wins the pot.
6. If both players choose to <PASS> or both players choose to <BET>, the player
↪  with the higher card wins the pot.

PLAYER INFORMATION:
You are player {Player ID}.

GAME HISTORY:
1. Blind ante: both player 0 and player 1 place 1 chip into the pot.
2. Deal: your card is shown in the image.
{Other History Information}

LEGAL ACTIONS:
<PASS>, <BET>.

INSTRUCTIONS:
Now it is your turn to choose an action. You should output your action in the
↪  following JSON format:
```json
{
    "action": "<ACTION>"
}
```
where <ACTION> is one of <PASS> and <BET>.

Do not include any extra commentary or explanation.
```

Back to competitive games.

## B.5  Atari Pong

**Visual observation.** An example is shown in Figure 8c. The two players each control a paddle on the side of the screen to hit a ball back and forth with each other. The paddles are vertical rectangles and the ball is a white square. The players score if the ball passes their opponent's paddle. The built-in bot controls the left paddle, while the VLM agent controls the right paddle. The scores of both players are displayed at the top of the screen.

**System prompt.**

```
You are an AI agent that maximizes your score in the game of Atari Pong.
```

**User prompt.**

```
GAME RULES:
1. Atari Pong is a zero-sum game played on a 2D screen with two players (left
↪ and right) and a ball.
2. Players each controls a paddle and receive rewards on different events:
    a. If the ball passes your paddle: the opponent +1 point.
    b. If the ball passes the opponent's paddle: you +1 point.
3. The ball bounces off the top/bottom walls and the paddles.
4. Paddles can only move vertically within the top and bottom walls.
5. First player to score 3 points wins.

PLAYER INFORMATION:
1. You are controlling the {Player Side} paddle.
2. The recent 4 game frames are given in chronological order, with the most
↪  recent frame at the end.
3. The ball is represented by a white square, and the paddles are represented by
↪  vertical rectangles.
4. Scores are displayed at the top of the screen.

LEGAL ACTIONS:
1. <UP>: move paddle upward.
2. <DOWN>: move paddle downward.
3. <STAY>: maintain current position (paddle has momentum, it stops gradually).

INSTRUCTIONS:
You should output your action in the following JSON format:
```json
{
    "action": "<ACTION>"
}
```
where <ACTION> is one of the following: <STAY>, <UP>, <DOWN>.
```

Back to competitive games.

### B.6  Coin's Dilemma

**Visual observation.** An example is shown in Figure 9. On the left of the image is a grid map showing

the current positions of all game elements, including two players (red and blue) and two coins (red

and blue). The players are each represented by a Pacman icon, and the coins are each represented

by a coin icon. On the right of the image is a table demonstrating the rewards of each event and a

corresponding counter tracking the number of occurrences for that event.

**System prompt.**

```
You are an AI agent that maximizes your score in the Coin Dilemma.
```

**User prompt.**

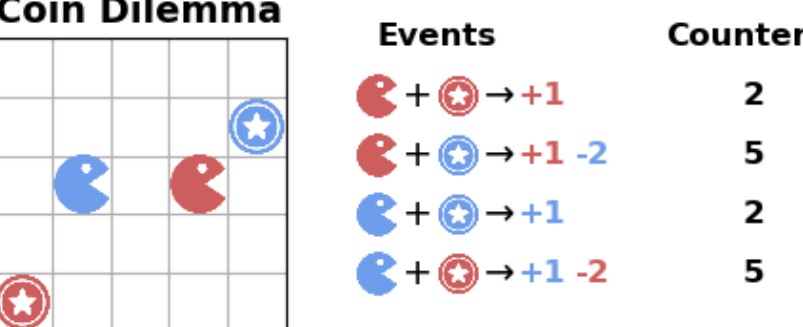

Figure 9: Example visual input of Coin Dilemma.

```
GAME RULES:
1. The Coin Dilemma is a general-sum game played on a 5x5 grid board with two
↪  players (red and blue) and two types of coins (red and blue).
2. Players receive rewards on different events:
    a. A player collects one coin of its own color: the player +1 point.
    b. A player collects one coin of the other player's color: the player +1
    ↪  point, the other player -2 points.
3. New coins spawn randomly on the board after each collection.

PLAYER INFORMATION:
1. You are the {Player Color} player.
2. The current game frame and a table of events and counters are shown in the
↪  image.
3. The red and blue players are represented by a red and blue pacman icon,
↪  respectively. The red and blue coins are represented by red and blue coin
↪  icons, respectively. If both players are in the same position, they are
↪  represented by a half-red-half-blue pacman icon.

LEGAL ACTIONS:
1. <UP>: move one step upward.
2. <DOWN>: move one step downward.
3. <LEFT>: move one step left.
4. <RIGHT>: move one step right.
5. <STAY>: stay in the current position.

INSTRUCTIONS:
You should output your action in the following JSON format:
```json
{
    "action": "<ACTION>"
}
```
where <ACTION> is one of the following: <STAY>, <RIGHT>, <LEFT>, <UP>, <DOWN>.
```

Back to mixed-motive games.

## B.7 Monster Hunt

**Visual observation.** An example is shown in Figure 10. On the left of the image is a grid map showing the current positions of all game elements, including two players (red and blue), two apples, and a monster. The players are each represented by a Pacman icon, the apples are each represented by a green apple icon, and the monster is represented by a black demon icon. On the right of the image is a table demonstrating the rewards of each event and a corresponding counter tracking the number of occurrences for that event.

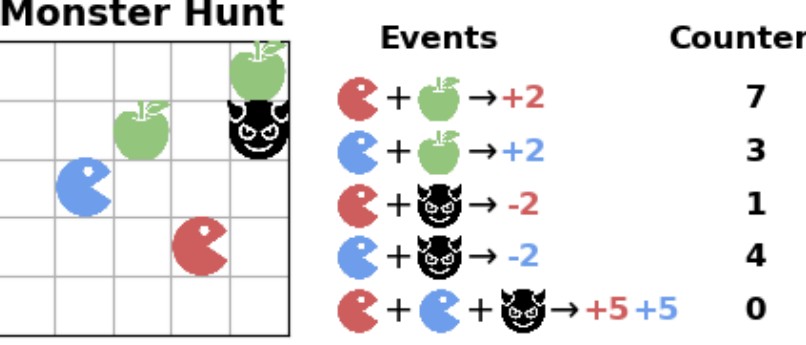

Figure 10: Example visual input of Monster Hunt.

**System prompt.**

> You are an AI agent that maximizes your score in the game of Monster Hunt.

**User prompt.**

```
GAME RULES:
1. Monster Hunt is a general-sum game played on a 5x5 grid board with two
↪  players (red and blue), one monster, and two apples.
2. The monster moves towards the closest player in each step.
3. Players move in the grid-world and receive rewards on different events:
     a. One player eats an apple: the player +2 points and the apple respawns at
     ↪  a random position.
     b. One player encounters the monster alone: the player -2 points and
     ↪  respawns at a random position.
     c. Two players defeat the monster together: both players +5 points and the
     ↪  monster respawns at a random position.

PLAYER INFORMATION:
1. You are the {Player Color} player.
2. The current game frame and a table of events and counters are shown in the
↪  image.
3. The red and blue players are represented by a red and blue pacman icon,
↪  respectively. The monster is represented by a black demon icon, and the
↪  apples are represented by green apple icons. If both players are in the same
↪  position, they are represented by a half-red-half-blue pacman icon.

LEGAL ACTIONS:
1. <UP>: move one step upward.
2. <DOWN>: move one step downward.
3. <LEFT>: move one step left.
4. <RIGHT>: move one step right.
5. <STAY>: stay in the current position.

INSTRUCTIONS:
You should output your action in the following JSON format:
```json
{
    "action": "<ACTION>"
}
```
where <ACTION> is one of the following: <STAY>, <RIGHT>, <LEFT>, <UP>, <DOWN>.
```

Back to mixed-motive games.

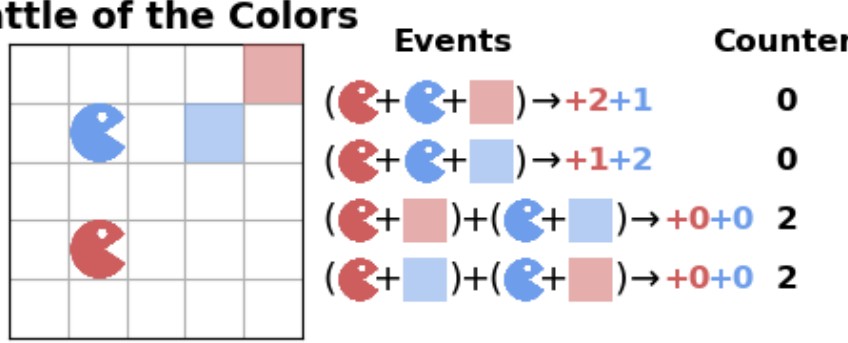

Figure 11: Example visual input of Battle of the Colors.

## B.8 Battle of the Colors

**Visual observation.** An example is shown in Figure 11. On the left of the image is a grid map showing the current positions of all game elements, including two players (red and blue) and two colored blocks (red and blue). The players are each represented by a Pacman icon. On the right of the image is a table demonstrating the rewards of each event and a corresponding counter tracking the number of occurrences for that event.

**System prompt.**

```
You are an AI agent that maximizes your score in the Battle of the Colors.
```

**User prompt.**

```
GAME RULES:
1. The Battle of the Colors is a general-sum game played on a 5x5 grid board
↪  with two players (red and blue) and two types of blocks (red and blue).
2. Players receive rewards on different events:
    a. When both players are on a red block: red player +2 points, blue player
    ↪  +1 point, and the red block will be refreshed to a new random position.
    b. When both players are on a blue block: red player +1 point, blue player
    ↪  +2 points, and the blue block will be refreshed to a new random
    ↪  position.
    c. When players are on different blocks: both players +0 points, and both
    ↪  blocks will be refreshed to new random positions.

PLAYER INFORMATION:
1. You are the {Player Color} player.
2. The current game frame and a table of events and counters are shown in the
↪  image.
3. The red and blue players are represented by red and blue pacman icons,
↪  respectively. The red and blue blocks are represented by red and blue
↪  rectangles, respectively. If both players are in the same position, they are
↪  represented by a half-red-half-blue pacman icon.

LEGAL ACTIONS:
1. <UP>: move one step upward.
2. <DOWN>: move one step downward.
3. <LEFT>: move one step left.
4. <RIGHT>: move one step right.
5. <STAY>: stay in the current position.
```

 **User prompt continued.**

```
INSTRUCTIONS:
You should output your action in the following JSON format:
```json
{
    "action": "<ACTION>"
}
```
where <ACTION> is one of the following: <STAY>, <RIGHT>, <LEFT>, <UP>, <DOWN>.

Do not include any extra commentary or explanation.
```

677

678 Back to mixed-motive games.

## C   Models configuration details

| Models | Version | Evaluated | Reasoning | Multimodal | Open-Source |
|---|---|---|---|---|---|
| o4-mini | o4-mini-2025-04-16 | ✓ | ✓ | ✓ | ✗ |
| claude-3-7-sonnet | claude-3-7-sonnet-20250219 | ✓ | ✓ | ✓ | ✗ |
| gemini-2.5-flash | gemini-2.5-flash-preview-04-17 | ✓ | ✓ | ✓ | ✗ |
| doubao-1-5-thinking-pro | doubao-1-5-thinking-pro-m-250415 | ✓ | ✓ | ✓ | ✗ |
| qvq-max | qvq-max-2025-03-25 | ✓ | ✓ | ✓ | ✗ |
| gpt-4.1 | gpt-4.1-2025-04-14 | ✓ | ✗ | ✓ | ✗ |
| claude-3-7 w/o thinking | claude-3-7-sonnet-20250219 | ✓ | ✗ | ✓ | ✗ |
| gemini-2.5 w/o thinking | gemini-2.5-flash-preview-04-17 | ✓ | ✗ | ✓ | ✗ |
| grok-2-vision | grok-2-vision-1212 | ✓ | ✗ | ✓ | ✗ |
| doubao-1-5-vision-pro | doubao-1-5-pro-32k-250115 | ✓ | ✗ | ✓ | ✗ |
| qwen-vl-max | qwen-vl-max-2025-04-08 | ✓ | ✗ | ✓ | ✗ |
| Llama-3.2-90B-Vision-Ins. | huggingface link | ✓ | ✗ | ✓ | ✓ |
| InternVL3-78B | huggingface link | ✓ | ✗ | ✓ | ✓ |
| Qwen2.5-VL-72B-Ins. | huggingface link | ✓ | ✗ | ✓ | ✓ |
| o3 | N/A | ✗ | ✓ | ✓ | ✗ |
| gemini-2.5-pro | N/A | ✗ | ✓ | ✓ | ✗ |
| grok-3 | N/A | ✗ | ✓ | ✗ | ✗ |
| deepseek-R1 | N/A | ✗ | ✓ | ✗ | ✓ |
| deepseek-V3 | N/A | ✗ | ✗ | ✗ | ✓ |

Table 3: Model configurations used in the evaluation.

680 The models used in our experiments are summarized in Table 3. For each model, the table specifies
681 the exact version, whether it possesses reasoning capabilities, supports multimodal inputs, and is
682 open-source. Note that for commercial models, we evaluated only the most recent versions released
683 before May 1, 2025, and did not assess any subsequent updates. For open-source models, the
684 corresponding repository URLs are recorded in the Version column. Because our benchmark is
685 designed specifically for VLMs, we excluded any models lacking multimodal input support (e.g.,
686 the DeepSeek and Grok-3 series). Additionally, we did not evaluate o3 (which requires a budget
687 exceeding $1000 to complete all tests) or `gemini-2.5-pro` (which is subject to a daily request limit
688 of 1000 requests per day).

689 Back to evaluation results of strategic reasoning and decision-making.

## D   Strategic reasoning evaluation details

691 Since we mainly use VLM APIs to conduct experiments, most experiments are run on personal
692 laptops. Experiments on open-source models are performed on an 8xA100 GPU server.

### D.1   Hanabi

694 We generated a high-quality dataset of 400 *Hanabi* game states to evaluate the strategic reasoning
695 performance of VLMs. First, 90% of the states were obtained through mutual prediction between two

reasoning models, while the remaining 10% were generated by a chat model forecasting a reasoning model's move. This approach exploits the relative weakness of chat models to sample game states that deviate more substantially from optimal play. For this study, we chose `doubao-1-5-thinking-pro` and `doubao-1-5-vision-pro`, both of which demonstrated top-tier decision-making performance, to represent the reasoning and chat model roles, respectively.

Second, in decision-making trials, `doubao-1-5-thinking-pro` selects *Play:Discard:Reveal* actions in a 2:3:4 ratio; we adopt this same distribution as the target action frequencies in our dataset.

Third, the dataset is balanced for player order—first and second players each account for 50% of the states—and the move index is uniformly distributed across the entire game sequence.

## D.2  Overcooked.

We generated a high-quality dataset of 400 *Overcooked* game states to evaluate the strategic reasoning performance of VLMs. Firstly, this dataset was derived from the Human Experiment Data provided by Overcooked-AI [12], which comprises game trajectories recorded from multiple human participants. The extraction focused on the trial-train subset of these data, with instances of invalid actions filtered out from the trajectories. Random sampling was then conducted on these filtered trajectories to ensure comprehensive coverage of possible game states. Each data instance comprises a sequence of four consecutive game frames.

Additionally, We applied constraints to simulate realistic game scenarios and control the distribution of target actions. Specifically, the proportion of the <stay> action among the target actions was limited to 10%. The dataset is balanced for two chefs, each accounting for 50% of the dataset.

## D.3  Breakthrough.

We generated a high-quality dataset of 400 *Breakthrough* game states to evaluate the strategic reasoning performance of VLMs. Each state was produced using a minimax algorithm with alpha–beta pruning, a widely adopted baseline in *Breakthrough* research. Since minimax search does not always reach terminal positions to determine win–loss outcomes, we implemented a state evaluation function: upon reaching a fixed search depth, we compute the difference between the maximum effective forward advancement of our deepest piece and that of the opponent's deepest piece, then normalize this difference to obtain a reward for the state. We configured minimax with maximum search depths for the first and second players as $(3, 4)$, $(3, 5)$, $(4, 5)$, $(4, 6)$, $(4, 4)$, and $(5, 5)$, respectively, and sampled move indices uniformly across the entire game sequence to ensure comprehensive coverage of possible game states.

## D.4  Kuhn poker.

We generated a high-quality dataset consisting of 400 *Kuhn Poker* game states to assess the strategic reasoning capabilities of VLMs. *Kuhn Poker* admits a mixed-strategy Nash equilibrium [32], characterized by a continuum of equilibrium strategies parameterized by a single probability $\alpha$, which denotes the likelihood of betting when holding a Jack. In our evaluation, we consider all pairwise matchups among three representative values of $\alpha$ (0, 1/6, and 1/3), resulting in nine distinct strategy combinations including self-play. For each combination, we simulate 600 head-to-head games and uniformly sample a total of 400 game states to construct the final dataset.

## D.5  Atari Pong

We generated a high-quality dataset consisting of 400 *Pong* game states to assess the strategic reasoning capabilities of VLMs. We uniformly sampled 400 state transitions from logged trajectories of two best-performing models in the decision-making process, namely `o4-mini` and `doubao-1-5-thinking-pro`, using the next actions of these VLM agents as ground truth. We then modified the prompts to ask VLMs to control the left paddle (the built-in bot's paddle) and predict those actions.

### D.6 Coin Dilemma.

We generated a high-quality dataset consisting of 400 *Coin Dilemma* game states to assess the strategic reasoning capabilities of VLMs. We consider two types of heuristic strategies for playing *Coin Dilemma* and generate the dataset by simulating game play with these strategies:

1. **Common Welfare**: player only collect the coin of its own color;
2. **Self Interest**: player will collect the closest coin, regardless of the color.

Concretely, we sample states from 6 settings, resulting in a dataset of 400 states:

1. **Common Welfare VS. Common Welfare**: sample 100 states;
2. **Self Interest VS. Self Interest**: sample 100 states;
3. **Common Welfare VS. Self Interest**: sample 50 states;
4. **Self Interest VS. Common Welfare**: sample 50 states;
5. **Random VS. Self Interest**: sample 50 states;
6. **Self Interest VS. Random**: sample 50 states;

For *Coin Dilemma*, we also record snapshots of the environment when sample the states for the dataset. This allows us to evaluate the action predictions from VLMs based on the outcome of such actions on the environment. The actions with the same outcome as the ground truth action are all considered correct. For example, if a player is at the top-left corner of the grid map, then action "UP" and "LEFT" are both considered correct with ground truth "STAY" as they all result in no movement of the player.

### D.7 Monster Hunt.

We generated a high-quality dataset consisting of 400 *Monster Hunt* game states to assess the strategic reasoning capabilities of VLMs. We consider four types of heuristic strategies for playing *Monster Hunt* and generate the dataset by simulating game play with these strategies:

1. **Common Welfare 1**: player will move directly towards the monster;
2. **Common Welfare 2**: player will move directly to the middle block of the grid map and stay there to wait for the other player and the monster;
3. **Common Welfare 3**: player will move directly to a certain corner of the grid map and stay there to wait for the other player and the monster;
4. **Self Interest**: player will move towards the closet apple.

Concretely, we sample states from 6 settings, resulting in a dataset of 400 states:

1. **Common Welfare 1 VS. Common Welfare 1**: sample 80 states;
2. **Common Welfare 2 VS. Common Welfare 2**: sample 80 states;
3. **Common Welfare 3 VS. Common Welfare 3**: sample 80 states;
4. **Self Interest VS. Self Interest**: sample 80 states;
5. **Random VS. Self Interest**: sample 40 states;
6. **Self Interest VS. Random**: sample 40 states;

For *Monster Hunt*, we also evaluate the action predictions based on the their outcomes, same as Coin Dilemma.

### D.8 Battle of the Colors.

We generated a high-quality dataset consisting of 400 *Battle of the Colors* game states to assess the strategic reasoning capabilities of VLMs. We consider four types of heuristic strategies for playing *Battle of the Colors* and generate the dataset by simulating game play with these strategies:

| Models | Cooperative | | | Competitive | | | | Mixed-Motive | | |
|---|---|---|---|---|---|---|---|---|---|---|
| | *Hanabi* return 1[3] | *Hanabi* return 2[4] | *Overcooked* | *Board* | *Poker* | *Pong* return 1[5] | *Pong* return 2[6] | *Dilemma* | *Hunt* | *Battle* |
| Optimal | 24.0 | 24.0 | 40.0 | 1.0 | 0.0 | 1.5 | 398.0 | 14.2 | 92.2 | 29.9 |
| gemini-2.5-flash | $6.5_{\pm 8.6}$ | $10.7_{\pm 5.3}$ | $3.6_{\pm 2.1}$ | $-0.6_{\pm 1.0}$ | $-0.1_{\pm 0.1}$ | $1.5_{\pm 0.0}$ | $194.4_{\pm 53.2}$ | $1.3_{\pm 4.8}$ | $15.6_{\pm 6.6}$ | $9.9_{\pm 2.9}$ |
| o4-mini | $10.3_{\pm 7.3}$ | $13.3_{\pm 2.9}$ | $7.0_{\pm 2.7}$ | $-0.4_{\pm 1.9}$ | $-0.1_{\pm 0.1}$ | $1.6_{\pm 0.2}$ | $205.2_{\pm 91.0}$ | $-0.8_{\pm 5.0}$ | $14.3_{\pm 10.5}$ | $1.2_{\pm 1.6}$ |
| doubao-1-5-thinking-pro | $13.6_{\pm 5.5}$ | $14.1_{\pm 4.0}$ | $4.2_{\pm 1.9}$ | $-0.8_{\pm 0.8}$ | $-0.1_{\pm 0.0}$ | $1.5_{\pm 0.0}$ | $230.5_{\pm 72.6}$ | $0.0_{\pm 1.0}$ | $6.8_{\pm 12.1}$ | $1.4_{\pm 1.4}$ |
| claude-3-7-sonnet | $1.6_{\pm 5.1}$ | $9.7_{\pm 3.9}$ | $4.2_{\pm 1.4}$ | $-0.6_{\pm 1.6}$ | $-0.1_{\pm 0.1}$ | $1.5_{\pm 0.0}$ | $133.7_{\pm 27.8}$ | $0.5_{\pm 3.4}$ | $9.4_{\pm 5.7}$ | $0.9_{\pm 1.4}$ |
| qvq-max | $0.0_{\pm 0.0}$ | $4.9_{\pm 2.9}$ | $1.0_{\pm 1.3}$ | $-0.9_{\pm 0.6}$ | $-0.2_{\pm 0.1}$ | $1.5_{\pm 0.0}$ | $158.2_{\pm 46.9}$ | $-0.1_{\pm 0.7}$ | $-9.4_{\pm 5.9}$ | $0.0_{\pm 0.0}$ |
| gemini-2.5 w/o thinking | $0.0_{\pm 0.0}$ | $3.8_{\pm 1.6}$ | $1.0_{\pm 1.6}$ | $-1.0_{\pm 0.0}$ | $-0.2_{\pm 0.1}$ | $1.5_{\pm 0.0}$ | $175.9_{\pm 41.5}$ | $-0.2_{\pm 1.5}$ | $-9.4_{\pm 10.2}$ | $0.9_{\pm 1.1}$ |
| gpt-4.1 | $0.0_{\pm 0.0}$ | $3.6_{\pm 1.4}$ | $0.0_{\pm 0.0}$ | $-1.0_{\pm 0.0}$ | $-0.3_{\pm 0.1}$ | $1.5_{\pm 0.0}$ | $151.8_{\pm 41.5}$ | $2.4_{\pm 2.7}$ | $0.9_{\pm 8.4}$ | $0.3_{\pm 0.6}$ |
| qwen-vl-max | $0.3_{\pm 0.5}$ | $0.3_{\pm 0.5}$ | $0.0_{\pm 0.0}$ | $-1.0_{\pm 0.0}$ | $-0.2_{\pm 0.0}$ | $1.5_{\pm 0.0}$ | $139.7_{\pm 29.3}$ | $-0.1_{\pm 0.7}$ | $2.9_{\pm 19.9}$ | $0.0_{\pm 0.0}$ |
| claude-3-7 w/o thinking | $0.0_{\pm 0.0}$ | $2.9_{\pm 0.9}$ | $1.0_{\pm 1.6}$ | $-0.9_{\pm 0.6}$ | $-0.4_{\pm 0.1}$ | $1.5_{\pm 0.0}$ | $121.4_{\pm 8.4}$ | $0.1_{\pm 2.7}$ | $-9.9_{\pm 9.1}$ | $0.5_{\pm 0.7}$ |
| grok-2-vision | $0.0_{\pm 0.0}$ | $1.6_{\pm 1.0}$ | $0.8_{\pm 1.3}$ | $-1.0_{\pm 0.0}$ | $-0.4_{\pm 0.1}$ | $1.6_{\pm 0.2}$ | $152.6_{\pm 45.9}$ | $0.1_{\pm 2.3}$ | $-10.5_{\pm 8.3}$ | $0.3_{\pm 0.6}$ |
| doubao-1-5-vision-pro | $0.0_{\pm 0.0}$ | $4.6_{\pm 1.1}$ | $0.0_{\pm 0.0}$ | $-1.0_{\pm 0.0}$ | $-0.4_{\pm 0.1}$ | $1.5_{\pm 0.0}$ | $121.4_{\pm 8.4}$ | $-0.4_{\pm 2.0}$ | $-2.4_{\pm 8.8}$ | $0.0_{\pm 0.0}$ |
| Qwen2.5-VL-72B-Ins. | $0.2_{\pm 0.4}$ | $0.2_{\pm 0.4}$ | $0.0_{\pm 0.0}$ | $-1.0_{\pm 0.0}$ | $-0.2_{\pm 0.1}$ | $1.5_{\pm 0.0}$ | $123.8_{\pm 4.6}$ | $-0.1_{\pm 0.9}$ | $9.1_{\pm 25.4}$ | $0.0_{\pm 0.0}$ |
| InternVL3-78B | $0.0_{\pm 0.0}$ | $2.4_{\pm 1.0}$ | $0.2_{\pm 0.6}$ | $-1.0_{\pm 0.0}$ | $-0.2_{\pm 0.1}$ | $1.5_{\pm 0.0}$ | $121.4_{\pm 8.4}$ | $0.8_{\pm 3.3}$ | $-11.9_{\pm 9.5}$ | $0.2_{\pm 0.5}$ |
| Llama-3.2-90B-Vision-Ins. | $0.0_{\pm 0.0}$ | $1.2_{\pm 1.6}$ | $0.8_{\pm 1.3}$ | $-1.0_{\pm 0.0}$ | $-0.3_{\pm 0.0}$ | $1.5_{\pm 0.0}$ | $121.4_{\pm 8.4}$ | $-0.1_{\pm 1.1}$ | $-6.6_{\pm 7.0}$ | $0.5_{\pm 0.7}$ |
| Random | 0.0 | 1.2 | 0.2 | $-1.0$ | $-0.5$ | 1.5 | 147.2 | $-0.1$ | $-10.1$ | 0.2 |

Table 4: Raw results for Decision-making.

1. **Common Welfare**: player will move to the closest color block (to both players) and stay there to wait for the other player;

2. **Self Interest**: player will move to the block of its own color.

3. **Biased Red**: player will move to the red block.

4. **Biased Blue**: player will move to the blue block.

Concretely, we sample states from 6 settings, resulting in a dataset of 400 states:

1. **Common Welfare VS. Common Welfare**: sample 100 states;

2. **Self Interest VS. Self Interest**: sample 100 states;

3. **Common Welfare VS. Self Interest**: sample 50 states;

4. **Self Interest VS. Common Welfare**: sample 50 states;

5. **Biased Red VS. Biased Red**: sample 50 states;

6. **Biased Blue VS. Biased Blue**: sample 50 states;

For *Battle of the Colors*, we also evaluate the action predictions based on the their outcomes, same as Coin Dilemma.

Back to evaluation results of strategic reasoning.

# E  Decision-making evaluation details

Since we mainly use VLM APIs to conduct experiments, most experiments are run on personal laptops. Experiments on open-source models are performed on an 8xA100 GPU server.

## E.1  Raw results without normalization

The raw data for Table 2 is presented in Table 4. Additionally, we present the fireworks reward for *Hanabi* and the step scores for *Pong*.

---

[3]*Hanabi* return 1 refers to the Final Reward mentioned in E.2

[4]*Hanabi* return 2 refers to the Fireworks Reward mentioned in E.2

[5]*Pong* return 1 refers to the Game Score mentioned in E.6

[6]*Pong* return 2 refers to the Step Score mentioned in E.6

### E.2 Hanabi

We employ two complementary evaluation metrics for the game *Hanabi*:

1. **Final Reward.** This metric, which is reported in the main text, assigns a score of 0 if all life tokens are consumed before the fireworks are completed, and a maximum of 25 if all fireworks stacks are built successfully. If neither terminal condition is reached earlier, play continues until the deck is exhausted plus one additional round. At game end, the values of the highest cards in each suit are summed to yield a total score out of 25.

2. **Fireworks Reward.** To relax the "zero-out" penalty upon losing all life tokens, this metric returns the partial fireworks progress at the moment the last life token is spent. Specifically, it computes the sum of the highest card values in each suit at that instant, rather than forcing a score of 0.

For each model under evaluation, we perform 10 self-play games and report the average *Final Reward* and *Fireworks Reward*. These results are then normalized and compared against a random baseline and an optimal policy derived from Independent PPO (IPPO) [84].

### E.3 Overcooked.

In *Overcooked*, each episode is limited to 50 timesteps. Within these timesteps, two chefs cooperatively cook soup and deliver the cooked soup to the service desk. The two chefs share a common cumulative points, where the final score for an episode is the sum of points accumulated at each timestep. The point obtained at each timestep is composed of two parts:

1. **Process-based point.** Awarded 2 points for specific beneficial actions, such as:

    (a) A chef successfully adds an onion to a cooking pot;

    (b) A chef picks up a dish when a pot contains onions or cooking is in progress;

    (c) A chef successfully plates a finished soup using a dish.

2. **Objective-based point.** Aligned with the game's goal of successfully delivering specified dishes, this reward is valued at 10 points upon successful delivery to the service desk.

For the 3-onion soup recipe, the total accumulated points for successfully completing and delivering one soup is 20 points, comprising the process-based and objective-based points. We evaluate each VLM through 10 episodes of self-play, where both chefs are controlled by the same type of model. We report the sum of the two chefs' cumulative points as the primary evaluation metric. We further normalize these scores with respect to the scores of the random policy and the optimal policy. The random policy uniformly samples actions at each step, while the optimal policy is defined as one that enables the two chefs to complete 2 full cooking-delivery processes within a single episode.

### E.4 Breakthrough.

In *Breakthrough*, we recorded the final outcomes by assigning a reward of $+1$ to the winner and $-1$ to the loser, as draws are not possible. We selected a moderately strong MCTS agent as our baseline, configured with an exploration constant $c = 2.0$, a maximum of 100 simulations per move, and a rollout count of 10. Each model played 20 games against this MCTS agent—10 as the first player and 10 as the second—and the mean outcome over all 20 games is reported. For the optimal policy, we employed a minimax agent with alpha–beta pruning and a maximum search depth of 5, using a state evaluation function as described in Section D.3. Although minimax is not guaranteed to be optimal for *Breakthrough*, it achieved a perfect win rate against the MCTS agent in our trials, making it a reasonable choice as the optimal policy in this study.

### E.5 Kuhn poker.

In *Kuhn Poker*, we measure the net chips won or lost by each player at the end of the game. Unlike online playing in other games, *Kuhn Poker* consists of only twelve information sets, each with two possible actions. For each VLM model, we estimated the policy by querying the model 25 times per information set and averaging the resulting action probabilities. We then calculated the exploitability

of the estimated policy, defined as the maximum expected loss against a best-response opponent. As the reference optimal strategies, we used the three mixed-strategy Nash equilibria described in Section D.4, each of which has zero exploitability.

## E.6 Atari Pong.

In Pong, players receive 1 point when the ball passes their opponents' paddle. We end the episode when one of the two players reach 3 points. We adopt frame stacking of 4 frames to pass dynamic information to the VLM agent. We also employ a sticky action probability of 0.25 and perform a random number (between 1 and 30) of "STAY" steps at the beginning of an episode to achieve randomness. These settings have been common practice in related works, such as DQN.

For evaluation, the VLMs all play against the same built-in bot from the game. As many VLMs fail to score even 1 point, the game scores themselves become too sparse for evaluating the performance of different models. We therefore design a denser metric that takes into account the number of steps that the VLM lasted against the bot. Specifically, the overall return is the addition of two parts, the score return and the step return:

$$
\begin{aligned}
R_{\text{all}} &= \tilde{R}_{\text{score}} + \tilde{R}_{\text{step}} \\
\tilde{R}_{\text{score}} &= \frac{R_{\text{score}}}{3.0} \times 90 \\
\tilde{R}_{\text{step}} &= \frac{N_{\text{step}} - N_{\text{min\_step}}}{N_{\text{max\_step}} - N_{\text{min\_step}}} \times 10
\end{aligned}
\tag{1}
$$

We evaluate each VLM for 10 episodes and report the mean and standard deviation of our designed score as the main metric.

## E.7 Coin Dilemma.

In Coin Dilemma, the players receive rewards on different game events:

1. **red player collects red coin**: red player +1 point;
2. **red player collects blue coin**: red player +1 point, blue player -2 points;
3. **blue player collects blue coin**: blue player +1 point;
4. **blue player collects red coin**: blue player +1 point, red player -2 points;

We evaluate each VLM through 10 episodes of self-play, where the red and blue players are controlled by the same type of model, and report the addition of two players' scores as the main metric. We further normalize these scores with respect to the scores of the random policy and the optimal policy. The random policy uniformly sample actions to take, while the optimal policy always moves directly towards the coin of the player's own color.

## E.8 Monster Hunt.

In Monster Hunt, the players receive rewards on different game events:

1. **red player collects apple**: red player +2 points;
2. **blue player collects apple**: blue player +2 points;
3. **red player encounters monster alone**: red player -2 points;
4. **blue player encounters monster alone**: blue player -2 points;
5. **both players defeat monster together**: both player +5 points;

We evaluate each VLM through 10 episodes of self-play, where the red and blue players are controlled by the same type of model, and report the addition of two players' scores as the main metric. We further normalize these scores with respect to the scores of the random policy and the optimal policy. The random policy uniformly sample actions to take, while the optimal policy always moves directly towards the middle block in the grid map and stay there to wait for the other player and the monster.

| Model | Hanabi | | | Board | | | Hunt | | |
|---|---|---|---|---|---|---|---|---|---|
| | text-only | multimodal | CoT | text-only | multimodal | CoT | text-only | multimodal | CoT |
| Optimal | 100.0 | 100.0 | 100.0 | 100.0 | 100.0 | 100.0 | 100.0 | 100.0 | 100.0 |
| gemini-2.5-flash | 42.0 | 37.0 | N/A | 23.5 | 23.2 | N/A | 16.5 | 32.0 | N/A |
| o4-mini | 53.8 | 58.2 | N/A | 27.5 | 26.8 | N/A | 18.0 | 36.2 | N/A |
| doubao-1-5-thinking-pro | 34.5 | 32.8 | N/A | 23.5 | 19.8 | N/A | 13.2 | 19.2 | N/A |
| claude-3-7-sonnet | 45.0 | 39.0 | N/A | 25.0 | 24.2 | N/A | 20.5 | 26.2 | N/A |
| qvq-max | 41.0 | 32.2 | N/A | 27.5 | 21.8 | N/A | 19.2 | 16.2 | N/A |
| gemini-2.5 w/o thinking | 24.5 | 21.5 | 24.0 | 20.5 | 14.8 | 21.5 | 12.5 | 23.0 | 13.5 |
| gpt-4.1 | 40.0 | 23.0 | 49.8 | 20.5 | 22.5 | 27.5 | 22.8 | 30.0 | 37.8 |
| qwen-vl-max | 17.0 | 26.5 | 20.0 | 19.0 | 19.5 | 17.2 | 17.0 | 23.5 | 22.5 |
| claude-3-7 w/o thinking | 19.2 | 9.8 | 32.8 | 19.2 | 18.0 | 19.0 | 31.2 | 25.8 | 25.8 |
| grok-2-vision | 23.8 | 12.8 | 22.5 | 14.0 | 10.8 | 18.2 | 12.0 | 22.0 | 28.2 |
| doubao-1-5-vision-pro | 19.5 | 15.0 | 25.2 | 17.2 | 15.8 | 16.8 | 13.8 | 32.2 | 26.2 |
| Qwen2.5-VL-72B-Ins. | 18.5 | 26.8 | 22.2 | 19.2 | 23.8 | 16.5 | 17.0 | 25.0 | 21.2 |
| InternVL3-78B | 26.8 | 25.2 | 20.5 | 17.5 | 14.0 | 16.0 | 23.5 | 23.2 | 23.2 |
| Llama-3.2-90B-Vision-Ins. | 26.8 | 20.0 | 14.8 | 6.5 | 11.8 | 14.0 | 18.2 | 23.5 | 19.5 |
| Random | 8.8 | 8.8 | 8.8 | 4.2 | 4.2 | 4.2 | 20.0 | 20.0 | 20.0 |

Table 5: All normalized results for Strategic reasoning.

### E.9 Battle of the Colors.

In Battle of the Colors, the players receive rewards on different game events:

1. **both players on red block**: red player +2 points, blue player +1 point;

2. **both players on blue block**: blue player +2 points, red player +1 point;

3. **players on different blocks**: both players +0 point;

We evaluate each VLM through 10 episodes of self-play, where the red and blue players are controlled by the same type of model, and report the addition of two players' scores as the main metric. We further normalize these scores with respect to the scores of the random policy and the optimal policy. The random policy uniformly sample actions to take, while the optimal policy always moves directly towards closest color block to the two players.

Back to evaluation results of decision-making.

## F  Additional experiment results

### F.1  Multimodal input results

Table 5 provides all normalized data for strategic reasoning. Specifically, we record the data obtained using multimodal input, text-only input, and CoT prompting. Table 6 provides all normalized data for decision making. Since reasoning models do not require CoT prompting, the corresponding entry is filled with N/A.

Back to analysis on multimodal input.

### F.2  Social behaviors results

The chat models demonstrate different behavior pattern compared to the reasoning models. For *Coin Dilemma*, as depicted in Figure 12a, the chat models shows inferior performance in collecting coins, resulting in less number of both cooperation and defections. The best performing chat model, GPT-4.1 achieves more number of cooperation over defections, indicating a behavior pattern that favor common welfare over self interest. The open source model InternVL3-78B, as depicted in Figure 12g, show a similar bahavior pattern to GPT-4.1. For *Monster Hunt*, as depicted in Figure 12b, the chat models fails to defeat the monster as often as the reasoning models overall. Among these chat models, only GPT-4.1 demonstrates a preference to collecting many apples, indicating self-interest-centered behavior, similar to many of the reasoning models. On the other hand, none of the open source models in Figure 12h exhibits this behavior pattern. For *Battle of the Color*, only gemini-2.5-flash is able to achieve considerable numbers of game events, demonstrating

| Model | Hanabi | | | Board | | | Hunt | | |
|---|---|---|---|---|---|---|---|---|---|
| | text-only | multimodal | CoT | text-only | multimodal | CoT | text-only | multimodal | CoT |
| Optimal | 100.0 | 100.0 | 100.0 | 100.0 | 100.0 | 100.0 | 100.0 | 100.0 | 100.0 |
| gemini-2.5-flash | $40.8_{\pm 21.9}$ | $27.1_{\pm 36.0}$ | N/A | $30.0_{\pm 84.5}$ | $20.0_{\pm 51.5}$ | N/A | $3.4_{\pm 12.8}$ | $26.2_{\pm 5.8}$ | N/A |
| o4-mini | $37.1_{\pm 26.1}$ | $42.9_{\pm 30.5}$ | N/A | $30.0_{\pm 94.0}$ | $30.0_{\pm 94.0}$ | N/A | $2.8_{\pm 8.4}$ | $24.9_{\pm 8.2}$ | N/A |
| doubao-1-5-thinking-pro | $37.5_{\pm 32.9}$ | $56.7_{\pm 22.8}$ | N/A | $15.0_{\pm 74.0}$ | $10.0_{\pm 42.0}$ | N/A | $13.5_{\pm 7.3}$ | $17.2_{\pm 11.3}$ | N/A |
| claude-3-7-sonnet | $33.8_{\pm 35.8}$ | $6.7_{\pm 21.1}$ | N/A | $45.0_{\pm 100.0}$ | $20.0_{\pm 79.5}$ | N/A | $11.8_{\pm 15.7}$ | $19.9_{\pm 3.5}$ | N/A |
| qvq-max | $0.0_{\pm 0.0}$ | $0.0_{\pm 0.0}$ | N/A | $5.0_{\pm 31.5}$ | $5.0_{\pm 31.5}$ | N/A | $9.4_{\pm 8.2}$ | $0.7_{\pm 4.5}$ | N/A |
| gemini-2.5 w/o thinking | $0.0_{\pm 0.0}$ | $0.0_{\pm 0.0}$ | $3.3_{\pm 10.5}$ | $0.0_{\pm 0.0}$ | $0.0_{\pm 0.0}$ | $20.0_{\pm 79.5}$ | $4.1_{\pm 8.6}$ | $0.7_{\pm 8.9}$ | $6.3_{\pm 9.8}$ |
| gpt-4.1 | $0.0_{\pm 0.0}$ | $0.0_{\pm 0.0}$ | $0.0_{\pm 0.0}$ | $0.0_{\pm 0.0}$ | $0.0_{\pm 0.0}$ | $10.0_{\pm 63.0}$ | $18.4_{\pm 15.6}$ | $11.2_{\pm 5.6}$ | $18.5_{\pm 10.9}$ |
| qwen-vl-max | $0.0_{\pm 0.0}$ | $1.2_{\pm 2.0}$ | $0.0_{\pm 0.0}$ | $5.0_{\pm 31.5}$ | $0.0_{\pm 0.0}$ | $0.0_{\pm 0.0}$ | $10.7_{\pm 14.7}$ | $13.2_{\pm 20.2}$ | $-0.6_{\pm 8.3}$ |
| claude-3-7 w/o thinking | $0.0_{\pm 0.0}$ | $0.0_{\pm 0.0}$ | $0.0_{\pm 0.0}$ | $5.0_{\pm 31.5}$ | $5.0_{\pm 31.5}$ | $10.0_{\pm 63.0}$ | $3.5_{\pm 6.9}$ | $0.2_{\pm 8.2}$ | $12.4_{\pm 8.6}$ |
| grok-2-vision | $0.0_{\pm 0.0}$ | $0.0_{\pm 0.0}$ | $0.0_{\pm 0.0}$ | $0.0_{\pm 0.0}$ | $0.0_{\pm 0.0}$ | $0.0_{\pm 0.0}$ | $0.9_{\pm 8.1}$ | $-0.4_{\pm 5.8}$ | $3.0_{\pm 3.9}$ |
| doubao-1-5-vision-pro | $0.0_{\pm 0.0}$ | $0.0_{\pm 0.0}$ | $5.0_{\pm 5.0}$ | $10.0_{\pm 42.0}$ | $0.0_{\pm 0.0}$ | $5.0_{\pm 31.5}$ | $7.8_{\pm 8.8}$ | $7.8_{\pm 8.2}$ | $16.2_{\pm 15.0}$ |
| Qwen2.5-VL-72B-Ins. | $6.2_{\pm 6.6}$ | $0.8_{\pm 1.8}$ | $2.9_{\pm 6.2}$ | $0.0_{\pm 0.0}$ | $0.0_{\pm 0.0}$ | $0.0_{\pm 0.0}$ | $9.1_{\pm 14.6}$ | $19.6_{\pm 25.7}$ | $23.3_{\pm 22.9}$ |
| InternVL3-78B | $0.0_{\pm 0.0}$ | $0.0_{\pm 0.0}$ | $1.7_{\pm 5.2}$ | $0.0_{\pm 0.0}$ | $0.0_{\pm 0.0}$ | $0.0_{\pm 0.0}$ | $5.6_{\pm 4.6}$ | $-1.8_{\pm 9.2}$ | $8.2_{\pm 7.6}$ |
| Llama-3.2-90B-Vision-Ins. | $0.0_{\pm 0.0}$ | $0.0_{\pm 0.0}$ | $0.0_{\pm 0.0}$ | $0.0_{\pm 0.0}$ | $0.0_{\pm 0.0}$ | $0.0_{\pm 0.0}$ | $-4.3_{\pm 6.2}$ | $3.6_{\pm 4.9}$ | $3.0_{\pm 8.8}$ |
| Random | 0.0 | 0.0 | 0.0 | 0.0 | 0.0 | 0.0 | 0.0 | 0.0 | 0.0 |

Table 6: All normalized results for Decision making.

superior reasoning and planing ability. From Figure 12f, `gemini-2.5-flash` demonstrates a strong willingness to cooperate with the other player, even if this means more points for the other player.

Back to analysis on social behaviors.

# G Failure case examples

## G.1 Strategic reasoning

We present three illustrative failure cases in strategic reasoning from different game environments as reference.

### G.1.1 Failure elixample in Hanabi

In *Hanabi*, each vision–language model (VLM) observes only the opponent's hand, creating a distinct information asymmetry. When predicting the opponent's next move, the VLM often immediately identifies which visible card could yield points and assumes the opponent will play it. This prediction, however, overlooks a critical consideration: from the opponent's perspective, their information about that card is incomplete, so they would not risk losing a life token by playing it prematurely. Instead, they would await more definitive clues before making that play. Detailed state information and the VLM's response are shown in Figure 13.

### G.1.2 Failure example in Overcooked

In *Overcooked*, accurately predicting the next action of the other player requires robust visual perception capabilities and a thorough comprehension of image information. Figure 14 illustrates a representative failure case stemming from shortcomings in these areas. In this instance, the VLM predicts the blue chef's action from the green chef's perspective. Despite correctly identifying from historical frames that the soup was cooked, the VLM's visual perception is inadequate; it fails to recognize that the blue chef was already holding the soup, plated in a dish. Simultaneously, the VLM overlooks a critical game rule, mistakenly believing that soup could be collected from the pot before a dish was acquired. As a result of these combined deficiencies in visual understanding and rule application, the VLM predicts the erroneous action <INTERACT>.

**VLM Response in this instance:**

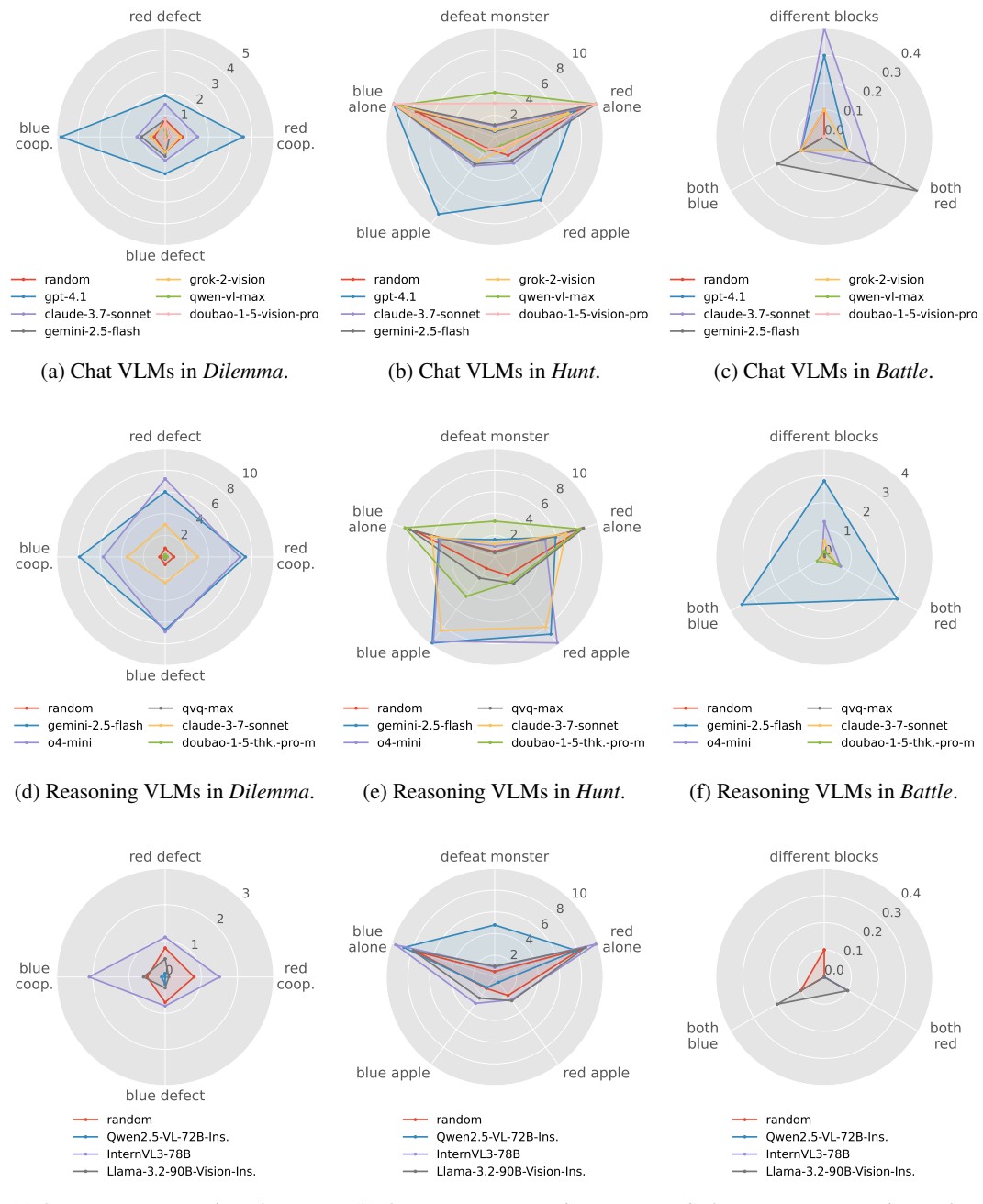

(a) Chat VLMs in *Dilemma*.   (b) Chat VLMs in *Hunt*.   (c) Chat VLMs in *Battle*.

(d) Reasoning VLMs in *Dilemma*.   (e) Reasoning VLMs in *Hunt*.   (f) Reasoning VLMs in *Battle*.

(g) Open-source VLMs in *Dilemma*.   (h) Open-source VLMs in *Hunt*.   (i) Open-source VLMs in *Battle*.

Figure 12: Behaviors of all models in mixed-motive social dilemma games. Dimensions are agents' behaviors described in Sec. 2.4.

```
......
Actually, let me look more carefully at what chef_0 is doing. In frames 23,
24, and 25, chef_0 is standing in the same position and hasn't moved. In
frame 26, the pot shows a black strip, indicating the soup is fully cooked.
According to the rules, after the soup is cooked for 5 timesteps, the next
step is to plate it using a dish. Since chef_0 is directly in front of the
pot, the most likely action would be <INTERACT> to take the soup from the
pot.
However, this would require chef_0 to have a dish already. Since I don't
clearly see chef_0 holding a dish, they would first need to get one.
Given this analysis, I believe chef_0 will most likely choose <INTERACT> to
interact with the pot.
......
```

33

**The Visualization of Failure State:**

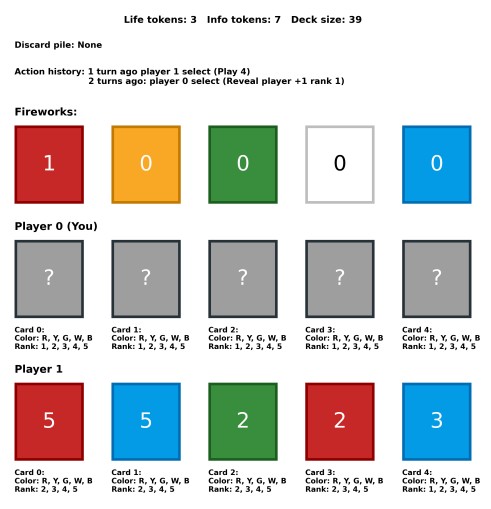

**VLM Response:**

```
......
Now, Player 1's possible actions:
play, discard, or reveal. Let's
consider playing first. The fireworks
for Red have 1, so the next needed is
Red 2. Player 1 has R2 (card 3), which
is the next in Red. Playing that would
be valid. Green needs 1 first, but
Player 1 has G2, which can't be played
yet since Green's 1 isn't down. The 5s
(R5, B5) can't be played until their
sequences are up to 4. B3 is Blue 3,
but Blue needs 1 first.
So the valid play here is R2 (card 3)
since Red is at 1, and R2 is the next.
Player 1, after knowing that card 3 is
rank 2 (from the reveal), and seeing
that Red needs 2, would likely play
that card.
......
```

Figure 13: Detailed information and VLM response for the Hanabi failure example.

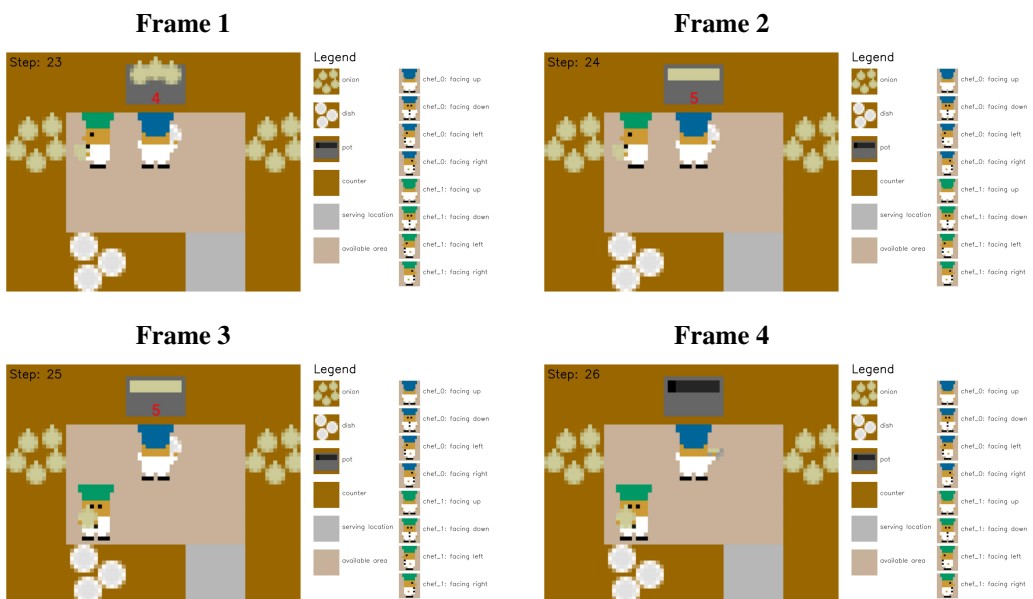

Figure 14: Failure example of reasoning in *Overcooked*.

### G.1.3 Failure example in Monster Hunt

In *Monster Hunt*, in order to predict the next action of the other player, it is important to track the positions and movements of all game elements, rather than focusing on only part of the game mechanics and make short-sighted predictions. A representative failure example is depicted in Figure 15, where the VLM notice the apple just one step above the red player, but does not pay attention to the monster on top of the apple. The VLM predicts the red player to go up to collect the apple, while the red player actually goes down to flee from the monster.

**The Visualization of Failure State:**     **VLM Response:**

Figure 15: Detailed information and VLM response for the Monster Hunt failure example.

**The Visualization of Failure State:**     **VLM Response:**

Figure 16: Detailed information and VLM response for the Breakthrough failure example.

## G.2  Decision-making

We further present three illustrative failure cases in decision-making from different game environments as reference too.

### G.2.1  Failure example in Breakthrough

In *Breakthrough*, a strong strategy must excel at both offense and defense. However, VLMs occasionally overlook situations in which the opponent's pieces are on the verge of victory and require immediate defensive intervention. Instead, they persistently advance their most distant piece by one step, thereby neglecting the critical balance between attack and defense.

### G.2.2  Failure example in Overcooked

In *Overcooked*, to correctly prepare a soup that needs three onion, the VLM agent needs to accurately manage the number of onions. A common failure mode of VLMs is that they often neglect these precise quantity requirements. As illustrated in Figure 17, the green chef initiates cooking by performing an <INTERACT> operation after adding only the second onion for this recipe. Concurrently, the blue chef prematurely begins to retrieve a plate. Consequently, the dish, even when served, fails to earn objective-based points because the ingredient count requirement was not met.

### G.2.3  Failure example in Pong

In *Pong*, in order to successfully intercept the ball, the VLM agent needs to accurately predict the landing point of the ball. A common failure mode of the VLMs is that they may over-adjust their

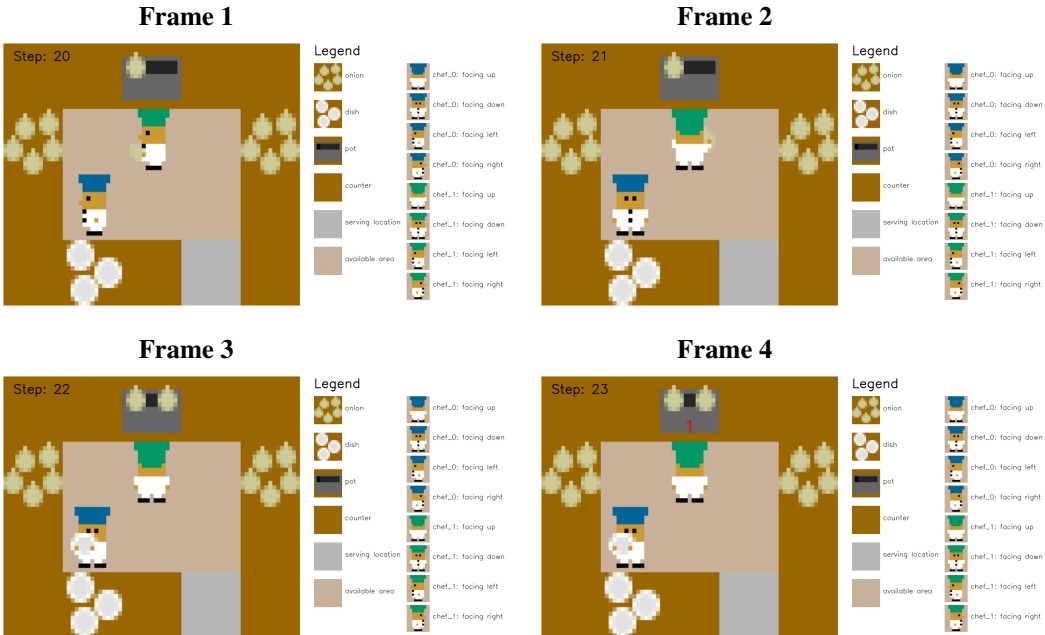

Figure 17: Failure example of decision-making in *Overcooked*.

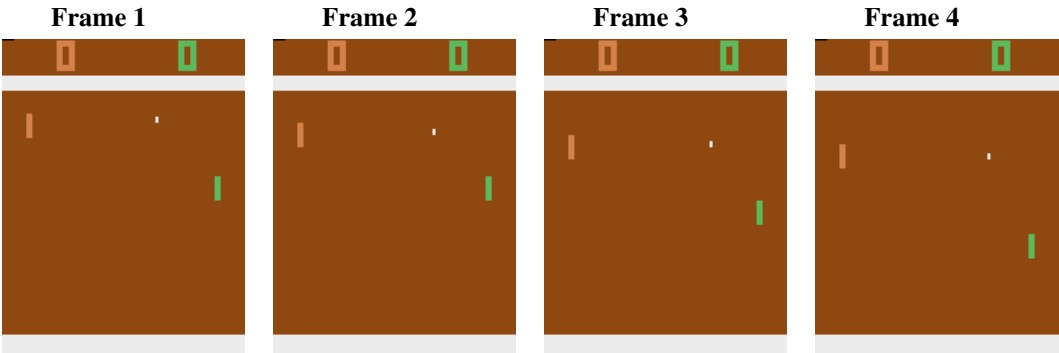

Figure 18: Failure example of decision-making in Pong.

paddle. As illustrated in Figure 18, the VLM on the right detects that the ball is moving downward and then move the paddle down, but misses the landing point due to over-aggressive adjustment.

## H Limitations

**Player number:** In real-world multi-agent scenarios, games often involve more than two participants. Although our current evaluation simplifies to two agents for tractability, some of our environments support more players (e.g., Hanabi support up to five players). Furthermore, our framework can be easily extended to other multi-player games.

**Human baseline:** At present, we compare VLMs only against random and optimal policies, lacking any human performance reference. A future improvement is to include human experiments with participants of varying expertise to establish a meaningful human baseline and clarify which level of human expertise the model approximates.

**Strategic reasoning evaluation:** Measuring strategic reasoning solely by prediction accuracy can overestimate performance when a model repeatedly selects the same action. Introducing metrics such as per-action precision, recall, and $F_1$ score will more comprehensively capture the model's reasoning ability.

**Decision-making: evaluation** Evaluating decision-making against a single opponent strategy does not test the VLM's full adaptability. Incorporating diverse baseline agents across all game scenarios will provide a more thorough assessment of their decision-making generalization and adaptability.

# I Broader impact

**Positive research and societal value.** VS-Bench targets a core capability that future AI systems will increasingly need: making strategic, multi-step decisions while perceiving the world through vision and language. By standardising how this ability is measured, the benchmark can accelerate reproducible research on safer, more reliable multimodal agents. Concretely, it enables (1) principled comparisons across models, and (2) diagnostic analyses that pinpoint specific failure modes such as myopic play or poor opponent modelling, and (3) a shared testbed for developing methods that foster cooperation, fairness, or robustness in complex interactive settings. Beyond academic progress, stronger decision-making agents could benefit applications like assistive household robotics, disaster-response swarms, automated traffic control, and large-scale scientific simulations where coordination and strategic planning are essential.

**Risk of misuse and dual-use considerations.** At the same time, more capable agents that reason strategically can be repurposed for adversarial or deceptive objectives — for example, collusive price-setting, automated disinformation campaigns, or the coordination of autonomous weapons systems. VS-Bench lowers the barrier to evaluating such capabilities, potentially making it easier to select or fine-tune models for harmful ends. To mitigate this, we (1) release only simulated environments that do not directly embody real-world attack surfaces, (2) distribute the benchmark and evaluation code under licenses that forbid the use of our assets in weaponised or surveillance applications, and (3) encourage follow-up work on safety safeguards (e.g., opponent-aware alignment checks) by providing explicit hooks for auditing model rationales and behaviours.

**Privacy and data ethics.** All VS-Bench environments are synthetic with no personally identifiable information or copyrighted third-party imagery that is not permissively licensed. Replays, logs, and intermediate states are derived entirely from simulation and are released under an open licence to avoids common privacy pitfalls in dataset creation and facilitates unrestricted academic use.

