# OpenReview forum: "VS-Bench: Evaluating VLMs for Strategic Reasoning and Decision-Making in Multi-Agent Environments"
_NeurIPS.cc/2025/Datasets_and_Benchmarks_Track — Submitted to NeurIPS 2025 Datasets and Benchmarks Track_

### Official Review · Reviewer_q2zX · 2025-06-30

**Rating:** 3
**Confidence:** 4

**Summary:**

The authors present VS-Bench, a new benchmark for testing how well VLMs can handle strategic, multi-agent games. They rightly point out that most current tests are for single agents or just text, so they've built a suite of eight vision-based games to cover cooperative, competitive, and mixed-motive scenarios. The core idea is to evaluate models in two ways: offline by predicting what another agent will do, and online by actually playing the games and seeing how well they perform. Their main finding, after testing a bunch of top models, is that current VLMs are still a long way from being expert players.

**Dataset Code Accessibility:**

Yes

**Ethical Considerations:**

No, there are no or only very minor ethics concerns

**Final Justification:**

The authors provided a very thorough rebuttal with significant new experiments, and I sincerely thank them for their efforts. After careful consideration of their response and the new data, I am maintaining my score of **3 (Borderline Reject)**.

Here is a breakdown of my reasoning:

### Resolved Issues:

The authors' rebuttal successfully addressed several of my key concerns, significantly strengthening the paper:

*   **Game Classification (W3):** My concern about the overly broad classification of games was completely resolved. The new table providing a detailed breakdown of game-theoretic properties and the specific abilities each game tests is an excellent addition and adds significant value.
*   **Agent Controllability (W4):** My point about the missed opportunity in prompting was also fully addressed. The new "persona" experiments are a fantastic contribution, providing valuable insights into the controllability and alignment of these models.
*   **Theory of Mind (W2 - Partially Resolved):** The qualitative example from Hanabi was a brilliant illustration of how VLMs can exhibit shallow reasoning. It added great color to the paper. However, it didn't fundamentally change the core evaluation methodology, which still relies on next-action prediction.

### Unresolved Issues:

My primary and most heavily-weighted concern remains unresolved:

*   **The "Visual" Necessity of the Benchmark (W1):** The central issue is that the benchmark doesn't seem to test for uniquely *visual* strategic reasoning. The game states are all fully describable in a symbolic format, meaning the visual interface is a convenience, not a necessity. The authors' new experiments, while well-executed, actually reinforced this concern for me. They show that models struggle with both (a) visual perception and (b) text-only strategic reasoning. This suggests the benchmark's main challenge is a combination of a noisy perception module feeding into a difficult symbolic reasoning task, rather than a single, integrated task of reasoning about inherently visual information.

### Weighting of these Aspects:

For a top-tier conference like NeurIPS, the novelty and soundness of a paper's core contribution are paramount. I have assigned a very high weight to the unresolved issue (W1) because it directly challenges the paper's central claim of being a *visual* strategy benchmark. While the improvements on W3 and W4 are excellent, they feel like valuable additions to a core premise that I still find shaky. In my view, the paper's main contribution is less about visual reasoning and more about showing how current VLMs fail when a symbolic reasoning task is presented through a visual front-end. This is an interesting finding, but it's different from the paper's primary framing.

---

### Short & Clear Points for the Authors:

*   Your work on the rebuttal was outstanding, and the new experiments on agent personas and the detailed game breakdown have made this a much stronger paper.
*   My main reason for the score is that the benchmark's tasks don't seem to *require* vision. The visual layer feels more like an optional, and often noisy, input channel for what are fundamentally symbolic games.
*   This weakens the paper's core claim of being a benchmark for *visual* strategic reasoning. The key difficulty seems to be the strategic reasoning itself, not the visual understanding.
*   I genuinely believe this work is of high quality. I would strongly encourage you to resubmit this work in the future, either by reframing it to more accurately reflect its contributions or by incorporating games where visual cues are truly indispensable to strategy.

**Limitations Weaknesses:**

While this is a great piece of work, I have a few concerns and suggestions where I think it could be made even stronger.

My main concern with the paper lies in how "visual" the benchmark truly is. While the goal is to move beyond text, many of the chosen games—like Hanabi, Breakthrough, or the grid worlds—feel like they could be easily and fully described with text. The visual interface is essentially just a rendering of a symbolic game state. This is somewhat backed up by the authors' own ablation study, where models sometimes did worse with images. It makes me question whether these tasks are the best testbed for uniquely visual strategic reasoning. The paper would be much stronger if it either included games where the visual information is truly indispensable, or dug deeper into why models fail to use the visual information that is provided.

Similarly, the evaluation of "theory of mind" feels a bit superficial. The paper uses next-action prediction as its main proxy for strategic reasoning, which is a fine start. But it only really tests for first-order ToM ("what will you do?"). It doesn't really get at the more complex, recursive reasoning that is key in games like Hanabi ("what do you think I know about my cards based on the hint I just gave you?"). A model could get decent prediction accuracy just by learning simple action patterns without truly reasoning about the other agent's mental state. It would be great to see the authors probe this more directly, perhaps by asking the VLM to explain the other agent's beliefs.

The classification of games into cooperative, competitive, and mixed-motive is also a little too neat. In reality, these skills are all tangled together. For instance, a cooperative game like Overcooked still requires you to model your teammate's intentions, which is a form of opponent modeling. It would be more insightful to have a more detailed breakdown of what specific capabilities (like long-term planning, risk assessment, etc.) each game tests for, rather than relying on these broad categories.

Finally, I think there's a bit of a missed opportunity when it comes to prompting. One of the most exciting things about VLM agents is that you can steer their behavior with instructions. The paper analyzes the emergent behaviors of the models in the social dilemma games, but what if you explicitly prompted them to be selfish, or cooperative, or to maximize the group's score? Testing how well the models adapt to these different "personas" would provide some really valuable insights into their controllability and alignment, which is a huge open question for agent research.

**Strengths Contributions:**

This is a strong, timely, and well-executed paper. The main contribution is tackling a clear gap in the literature: evaluating VLMs in multi-agent, multimodal settings. I was impressed by the benchmark's design. The choice of games is smart, covering the classic cooperative, competitive, and mixed-motive setups from game theory, which gives a broad test of strategic skills. I particularly liked the dual-evaluation approach—separating offline strategic reasoning (predicting moves) from online decision-making (playing the game). This gives a much clearer picture of where models are failing. The empirical work is thorough, testing a wide range of recent models, and the paper itself is very well-written and easy to follow. The authors have clearly put a lot of effort into the details and reproducibility, which is a huge plus.

---

> ### Author Rebuttal · Authors · 2025-07-31
>
> Thank you for your constructive comments and questions! We hope the following response can address your concerns.
>
> ---
>
> > W1: My main concern with the paper lies in how "visual" the benchmark truly is. ... The paper would be much stronger if it either included games where the visual information is truly indispensable, or dug deeper into why models fail to use the visual information that is provided.
>
> We agree with the reviewer that the state of some games in our benchmark can be fully described with text. However, we would like to argue that visual observation is a more natural representation than text observation in these games. For example, in board games like Tic-Tac-Toe, we can represent the state with
> 1. An image of the board.
> 2. A text string like "OX_\n_X_\nXO_".
>
> The text string transforms the 2D board into a 1D string, which makes the spatio concept of "vertical" harder to understand, as it requires the models to correlate multiple tokens separated by a specific distance in the 1D input. The same applies to other board games and gridworld games in our benchmark. Therefore, in principle, the 2D visual observation should be an easier and more natural way for models to percept and should lead to better results.
>
> However, our own ablation study shows that existing models sometimes did worse with images than with text. We attribute this to current models being stronger in the text modality than in the visual modality: most released models first achieve gains in textual reasoning before improving on visual reasoning tasks. The discrepancy between our ablation results and the ideal outcome underscores the limitations in visual reasoning and decision-making of existing models: they ought to perform at least no worse than on text, but they fail to do so. We believe revealing such a deficiency of existing models is one of the values of our benchmark.
>
> To understand why models fail to use the visual observations to make decisions, we perform 2 additional experiments: (1) visual perception and (2) text-only decision-making. The conclusion is that the deficiency comes from the coupled difficulty of visual perception and strategic decision-making.
>
> (1) Perception: we evaluate the perception ability of VLMs by asking them to output a dict or a matrix representation of all information in the image (board state, card number, etc) and report their accuracy. *Pong* is not evaluated because its states are pixel-based and hard to represent by text.
>
> ||Overall|*Hanabi*|*Overcooked*|*Breakthrough*|*Poker*|*Dilemma*|*Hunt*|*Battle*|
> |:-:|:-:|:-:|:-:|:-:|:-:|:-:|:-:|:-:|
> |`gemini-2.5-flash`|80.3|79.9|54.5|98.5|100.0|76.4|73.0|79.9
> |`o4-mini`|85.3|79.7|69.8|97.2|99.1|85.5|80.2|85.4|
> |`doubao-1-5-thinking-pro`|73.7|42.2|52.4|91.0|98.6|76.7|74.9|80.0|
> |`claude-3-7-sonnet`|79.8|73.0|62.8|75.2|99.5|82.6|79.7|85.7|
> |`qvq-max`|76.6|75.1|63.6|83.3|95.2|69.5|72.0|77.7|
> |`gemini-2.5` w/o thinking|82.9|79.9|38.8|88.2|97.2|92.5|93.1|90.8|
> |`gpt-4.1`|76.5|72.1|62.0|67.0|100.0|76.7|76.8|81.2|
> |`qwen-vl-max`|80.0|76.1|68.2|81.2|99.2|78.4|76.4|80.6|
> |`claude-3-7` w/o thinking|79.9|75.9|59.7|79.0|99.6|81.8|80.4|82.8|
> |`grok-2-vision`|70.3|75.2|46.8|80.3|59.5|76.4|73.3|81.0|
> |`doubao-1-5-vision-pro`|77.0|80.0|33.1|89.3|100.0|78.0|77.6|81.2|
> |`Qwen2.5-VL-72B-Ins.`|80.7|76.0|72.9|75.1|100.0|79.8|79.0|82.4|
> |`InternVL3-78B`|74.5|74.6|43.6|64.3|99.2|81.1|76.7|81.8|
> |`Llama-3.2-90B-Vision-Ins`|62.5|30.7|58.6|59.7|81.6|68.5|66.0|72.4|
>
> In general, existing VLMs achieve a good performance of 70-80% perception accuracy in visual perception. However, they may miss some detailed information that is critical for decision-making. For example, in *Overcooked*, VLMs are not good at recognizing the players' facing direction, making it difficult to determine whether to interact with the facing object.
>
> (2) Text-only decision-making: we evaluate the decision-making ability of VLMs with text-only observations.
>
> ||Overall|*Hanabi*|*Overcooked*|*Board*|*Poker*|*Dilemma*|*Hunt*|*Battle*|
> |:-:|:-:|:-:|:-:|:-:|:-:|:-:|:-:|:-:|
> |`gemini-2.5-flash`|22.5|40.8 (21.9)|61.3 (19.7)|30.0 (42.2)|28.2 (60.9)|-26.3 (18.1)|3.4 (12.8)|20.2 (5.7)|
> |`o4-mini`|27.1|37.1 (26.1)|58.8 (23.3)|30.0 (47.0)|79.1 (33.6)|-42.7 (22.4)|2.8 (8.4)|24.8 (13.7)|
> |`doubao-1-5-thinking-pro`|29.7|37.5 (32.9)|80.9 (30.6)|15.0 (37.0)|62.7 (41.8)|-22.8 (19.6)|13.5 (7.3)|21.2 (7.2)|
> |`claude-3-7-sonnet`|35.9|33.8 (35.8)|81.9 (26.8)|45.0 (50.0)|76.4 (42.7)|-37.4 (18.4)|11.8 (15.7)|39.7 (10.4)|
> |`qvq-max`|3.2|0.0 (0.0)|23.1 (13.1)|5.0 (15.8)|-22.7 (55.5)|-1.8 (12.2)|9.4 (8.2)|9.6 (8.1)|
> |`gemini-2.5` w/o thinking|4.8|0.0 (0.0)|1.0 (2.3)|0.0 (0.0)|2.7 (10.9)|23.8 (13.3)|4.1 (8.6)|2.0 (3.4)|
> |`gpt-4.1`|11.7|0.0 (0.0)|0.0 (1.5)|0.0 (0.0)|1.8 (38.2)|40.2 (12.0)|18.4 (15.6)|21.7 (12.4)|
> |`qwen-vl-max`|3.2|0.0 (0.0)|1.0 (2.3)|5.0 (15.8)|1.8 (60.9)|3.6 (5.0)|10.7 (14.7)|0.0 (1.5)|
> |`claude-3-7` w/o thinking|9.6|0.0 (0.0)|4.5 (5.5)|5.0 (15.8)|0.9 (23.6)|35.6 (18.1)|3.5 (6.9)|17.7 (10.2)|
> |`grok-2-vision`|1.8|0.0 (0.0)|0.0 (1.5)|0.0 (0.0)|10.9 (55.5)|-1.1 (1.8)|0.9 (8.1)|1.5 (2.5)|
> |`doubao-1-5-vision-pro`|1.3|0.0 (0.0)|-0.5 (0.0)|10.0 (21.0)|-13.6 (24.6)|5.3 (13.5)|7.8 (8.8)|0.0 (1.5)|
> |`Qwen2.5-VL-72B-Ins.`|2.9|6.2 (6.6)|-0.5 (0.0)|0.0 (0.0)|2.7 (36.4)|2.1 (3.6)|9.1 (14.6)|0.5 (2.0)|
> |`InternVL3-78B`|0.8|0.0 (0.0)|3.0 (3.9)|0.0 (0.0)|-7.3 (45.5)|3.2 (3.0)|5.6 (4.6)|1.2 (2.0)|
> |`Llama-3.2-90B-Vision-Ins`|22.5|40.8 (21.9)|61.3 (19.7)|30.0 (42.2)|28.2 (60.9)|-26.3 (18.1)|3.4 (12.8)|20.2 (5.7)|
>
> In general, without visual perception, existing models still struggle in decision-making, achieving **<36%** normalized return. VLMs sometimes are better in text-only settings, and sometimes are better with visual observation.
>
> > W2: Similarly, the evaluation of "theory of mind" feels a bit superficial. ... It would be great to see the authors probe this more directly, perhaps by asking the VLM to explain the other agent's beliefs.
>
> We agree with the reviewer that our next-action prediction accuracy mainly tests for first-order ToM. We use it as the main metric for strategic reasoning because it is a general metric that can be calculated in all environments and provides quantitative results.
>
> To further provide an in-depth qualitative evaluation, we follow the reviewer's suggestion to probe the ToM ability by asking the VLM to explain the other agent's belief. We consider a typical case in *Hanabi*: player 0 has two Yellow-2 that are playable, but they have no information about these cards. Player 1 chooses to reveal the rank 2 to player 0, hinting to player 0 to play one of the cards. We ask player 1 to explain player 0's belief, and the response is
>
> ```
> "Why did the partner spend a token just to tell me I have two 2-cards?" A rank-only clue is rarely given unless at least one of the pointed cards is immediately useful. Right now, the only colours that still need a 2 are Yellow and White (the Red, Green, and Blue 2s are already on the table). Therefore, the partner is practically telling me: "At least one of those cards is the Yellow-2 or the White-2 and is playable right now." So I should play one card.
> ```
>
> However, player 1 fails to notice they currently have only 1 life token and would lose the game if they play one more wrong card. So player 0's actual reasoning is "I'm not sure about these cards and cannot risk playing a wrong card" and chooses to discard a card. This qualitative case study validates that VLMs show some level of ToM reasoning, but can fail to accurately infer other agents' beliefs, which aligns with our evaluation results using next-action prediction accuracy.
>
>
> > W3: The classification of games into cooperative, competitive, and mixed-motive is also a little too neat. ...
>
> We provide a detailed Breakdown of our games, considering
> * Game-theoretic properties: interaction type, full/partial observation, deterministic/probabilistic transition, simultaneous/sequential move.
> * Multi-agent abilities: spatial reasoning, numerical reasoning, opponent modelling, long-term planning, risk assessment, and team collaboration.
>
> ||Interaction|Full Obs.|Deterministic|Simultaneous|Spatial Reasoning|Numerical Reasoning|Opponent Modeling|Long-Term Planning|Risk Assessment|Team Collaboration|
> |:-:|:-:|:-:|:-:|:-:|:-:|:-:|:-:|:-:|:-:|:-:|
> |*Hanabi*|cooperative|✗|✗|✗|low|middle|high|high|middle|high|
> |*Overcooked*|cooperative|✓|✓|✓|high|low|middle|middle|low|high|
> |*Breakthrough*|competitive|✓|✓|✗|high|low|middle|high|high|low|
> |*Poker*|competitive|✗|✗|✗|low|high|middle|low|high|low|
> |*Pong*|competitive|✓|✗|✓|high|low|low|low|middle|low|
> |*Dilemma*|mixed-motive|✓|✗|✓|middle|middle|high|middle|middle|middle|
> |*Hunt*|mixed-motive|✓|✗|✓|middle|middle|high|middle|high|middle|
> |*Battle*|mixed-motive|✓|✗|✓|middle|middle|high|middle|middle|middle|
>
> We will add this table in our revised paper.
>
> > W4: Finally, I think there's a bit of a missed opportunity when it comes to prompting. ...
>
> We perform additional experiments on "personas" in 3 mixed-motive games. We select the top 3 models and prompt them to be selfish (maximize their own return) and cooperative (maximize the group's total return).
>
> ||`o4-mini`|`gemini-2.5-flash`|`doubao-1-5-thinking-pro`|
> |:-:|:-:|:-:|:-:|
> |*Dilemma* cooperative|72.8 (11.1)|53.7 (10.4)|15.7 (7.1)|
> |*Dilemma* selfish|-21.4 (16.1)|4.3 (10.3)|-5.3 (14.2)|
> |*Hunt* cooperative|57.1 (22.7)|15.9 (16.4)|15.7 (27.7)|
> |*Hunt* selfish|19.1 (4.0)|16.8 (5.0)|11.3 (3.8)|
> |*Battle* cooperative|19.2 (10.5)|22.2 (9.1)|6.1 (6.0)|
> |*Battle* selfish|4.0 (6.2)|2.5 (4.6)|0.0 (1.5)|
>
> The results in the table show that the top 3 VLMs follow the instructed "personas" and produce corresponding behaviors and results. For example, in *Coin Dilemma*, cooperative agents achieve win-win high returns, while selfish agents perform worse than random.
>
> ---
>
> We genuinely value your dedication to reviewing our paper. We have carefully addressed each of your concerns, and we sincerely hope our efforts merit a raise in your rating.

---

> > ### Comment · Reviewer_q2zX · 2025-08-02
> >
> > Thank you so much for the incredibly detailed and thoughtful rebuttal. I genuinely appreciate the amount of work you've put into the additional experiments and analyses. It has been very helpful in clarifying my thoughts on the paper and has significantly strengthened the submission in several key areas.
> >
> > I want to start by saying you've completely addressed some of my concerns. The new table breaking down the capabilities tested by each game (W3) is fantastic—it's exactly the kind of detailed analysis I was hoping for and adds a lot of depth. Similarly, the new "persona" experiments (W4) are a great addition. Testing how models react to explicit prompts for selfish or cooperative behavior provides a valuable new layer of insight into their controllability, which is a really important contribution.
> >
> > On the Theory of Mind point (W2), I have to say, the Hanabi case study was a perfect example and brilliantly illustrated your point. It's a fantastic qualitative insight showing that VLMs can produce plausible-looking reasoning that is ultimately shallow and misses critical context. My concern here, however, was less about finding a single compelling case and more about the evaluation methodology itself. While the anecdote is powerful, it doesn't change the fact that the benchmark's primary metric for ToM remains next-action prediction, which we both agree is a bit superficial. This is a minor point, but I thought it was worth mentioning.
> >
> > This brings me to my main concern, W1, regarding how "visual" the benchmark truly is. I've read your arguments carefully, and while I understand your point that vision is a more "natural" 2D representation than a 1D text string, I feel this doesn't quite resolve the core issue I was trying to raise. My worry wasn't about "naturalness," but about "necessity."
> >
> > Frankly, your new experiments, while very well-executed, seem to have reinforced my initial worry. Here's how I see it:
> >
> > Your perception experiment shows that even the best models only achieve 70-80% accuracy in translating the image to a symbolic state. This means the visual channel is inherently noisy and lossy.
> > Your text-only experiment shows that even with a perfect, lossless symbolic state, the models still struggle badly with the strategic decision-making part of the task.
> > At the end of the day, what this tells me is that the primary bottleneck is the strategic reasoning, and the visual front-end acts as an additional layer of difficulty by introducing perceptual errors before the model even gets to the core problem. So, the benchmark isn't really testing for uniquely visual strategic reasoning (i.e., where the visual information itself is key to the strategy in a way that text can't capture). Instead, it's testing a model's ability to perform symbolic reasoning after surviving a noisy visual perception stage. While this is a valid thing to test, it feels different from the paper's main framing and, in my view, slightly lessens the novelty of the "visual" contribution.
> >
> > This is the kind of foundational issue that, in my opinion, can't really be patched during a short rebuttal period. It's more about the core framing and design of the benchmark itself.
> >
> > So, to be very clear: this is a strong, valuable, and well-executed piece of work. The effort you've shown in the rebuttal is exemplary. However, because of this lingering concern about the core contribution—what the "V" in VS-Bench is truly testing—I'm afraid I have to stand by my initial rating. This is a classic "borderline" paper for me; it's technically solid and has many strengths, but the core premise has a weakness that's hard to overlook for a top-tier venue like NeurIPS.
> >
> > I genuinely believe that with some reframing of the paper's contribution, or by adding a game or two where the visual component is truly indispensable, this would be a clear accept. I truly hope you'll continue this line of work.
> >
> > Thanks again for the detailed response and the engaging discussion. I look forward to seeing the next version of this work.

---

> > ### Author Response · Authors · 2025-08-02
> > **Two Additional Visual-Indispensable Environments to Strengthen W1**
> >
> > Thank you for your response! We are encouraged that our rebuttal addressed your concern on W2 (with minor comments), W3, and W4.
> >
> > For your constructive suggestions on W1, **we are dedicated to adding 2 more environments where the visual component is truly indispensable and will report the experiment results as soon as they are ready.** While we are working on it, we would like to briefly discuss the 2 additional environments and 1 existing environment, and explain why the visual component is indispensable in each.
> >
> > *Knights Archers Zombies (KAZ) from PettingZoo*
> >
> > * Environment description: a cooperative video game where nights and archers defend the bottom of the screen against zombies that descend along unpredictable paths. Please see the [official documentation](https://pettingzoo.farama.org/environments/butterfly/knights_archers_zombies/) for visualization.
> > * Why the visual component is indispensable: agents must track continuous, time-varying geometry (headings, melee arcs, arrow trajectories, irregular zombie motion) and react in real time—information, which cannot be faithfully or compactly represented as discrete text without losing critical information for decision-making.
> >
> > *Simple Push from MPE*
> >
> > * Environment description: a competitive MPE task with one good agent, one adversary, and one landmark: the good agent is rewarded for being close to the landmark, while the adversary is rewarded for being near the landmark and pushing the agent away. Please see the [official documentation](https://pettingzoo.farama.org/environments/mpe/simple_push/) for visualization.
> > * Why the visual component is indispensable: agents must perceive fine-grained, continuous contact geometry—exact relative pose, approach angles, micro-collisions, and brief occlusions—that evolve each timestep; encoding these cues as text would lose timing and precision, whereas pixels compactly preserve them for real-time control.
> >
> > *Existing Environment: Atari Pong*
> > * Why the visual component is indispensable: All task-critical cues—ball/paddle positions and velocities, bounce angles off walls/paddles, and serve timing—are only provided as high-frequency pixel dynamics, so expressing them as text would lose timing and precision.
> >
> > We would be grateful if you could let us know if these environments meet your expectations for demonstrating "visual component is truly indispensable". Your suggestions and confirmation are invaluable to us.

---

> > > ### Author Response · Authors · 2025-08-05
> > > **Results on Additional Visual-Indispensable Environments**
> > >
> > > Thank you again for your constructive feedback! As discussed, we have added two new environments including Knights Archers Zombies (KAZ) and Simple Push where the visual component is truly indispensable, along with retaining Atari Pong as an existing example. We have completed the decision-making evaluations and will add the following results to our revised paper.
> > >
> > > |  | `gemini-2.5-flash` | `o4-mini` | `doubao-1.5-thinking-pro` | `claude-3.7-sonnet` | `qvq-max` | `gemini-2.5` w/o thinking | `gpt-4.1` | `qwen-vl-max` | `claude-3-7` w/o thinking | `grok-2` | `doubao-1.5-vision-pro` | `Qwen2.5-VL-72B-Ins.` | `InternVL3-78B` | `Llama-3.2-90B-Vision-Ins.` |
> > > |:-:|:-:|:-:|:-:|:-:|:-:|:-:|:-:|:-:|:-:|:-:|:-:|:-:|:-:|:-:|
> > > | KAZ | 14.1 (9.0) | 14.1 (13.1) | 14.1 (9.0) | 5.3 (7.4) | -1.7 (5.0) | 0.0 (4.3) | -5.3 (0.0) | -5.3 (0.0) | 3.5 (9.0) | 0.0 (7.4) | -5.3 (0.0) | -5.3 (0.0) | -3.5 (2.5) | 0.0 (4.3) |
> > > | Simple Push | 32.7 (33.8) | 36.9 (31.4) | 38.5 (39.0) | 39.4 (32.3) | 38.6 (31.6) | 23.8 (29.4) | 31.5 (25.1) | 14.4 (45.5) | 5.7 (53.2) | -58.2 (94.0) | 32.9 (35.4) | 19.7 (39.9) | 5.9 (56.1) | -29.6 (66.9) |
> > >
> > > These additional results align with the findings in our manuscript that even SOTA models struggle at decision-making in multi-agent games. Moreover, these environments feature critical and fast-changing spatio cues that are only available in visual observations rather than text.
> > >
> > > We hope these results address your concern by demonstrating that **VS-Bench now evaluates 3 environments where visual perception is not merely a lossy preprocessing stage, but an indispensable part of the strategic decision-making process**. We sincerely appreciate the time and effort you dedicated to reviewing our submission and would be very grateful if you could consider re-evaluating our paper based on these results.

---

> > > > ### Comment · Reviewer_q2zX · 2025-08-05
> > > >
> > > > I am genuinely impressed and want to start by thanking you for the incredible amount of work you've put in over the last few days. Running entirely new experiments on new environments during the rebuttal period is a huge effort, and it speaks volumes about your dedication to this research. I sincerely appreciate it.
> > > >
> > > > Let me be clear: the new environments you've proposed and tested—KAZ and Simple Push—are exactly the kind of examples I had in mind when I talked about "visual indispensability." You've perfectly understood and addressed the core of my suggestion on that front. The results are interesting and absolutely strengthen the benchmark.
> > > >
> > > > This brings me to a difficult point, and it's one that's more about the constraints of the academic review process than the quality of your additions. My original review, and my subsequent rating, was based on an evaluation of the paper as a whole—the narrative, the analysis, and the conclusions you drew from the initial set of eight environments. My primary concern was that the paper's central claim about being a visual benchmark felt disconnected from the fact that most of the environments were, fundamentally, symbolic puzzles with a visual interface.
> > > >
> > > > By adding these two new, truly visual environments, you are not just adding a couple of rows to a results table. You are fundamentally altering the composition and the core argument of your paper. A proper evaluation would require me to see how this new evidence is woven into the entire manuscript. For instance, how does this change the paper's main narrative in the introduction and abstract? How do you analyze these results in relation to the others? Does this shift the overall conclusion from "VLMs are bad at strategic reasoning" to a more nuanced take like, "VLMs fail at symbolic games due to noisy perception, but fail at visual games for different reasons"?
> > > >
> > > > These are not minor edits; they represent a significant revision of the paper's logic and structure. Since I can't see a revised manuscript at this stage, I'm being asked to evaluate the promise of a future revision based on a table of new results. I don't feel I can do that fairly. I need to assess the final, coherent work in its entirety.
> > > >
> > > > So, to be transparent, while my appreciation for your work and your responsiveness has grown immensely, I feel I have to stand by my original rating. This decision is not a reflection on the quality of your new results, which are excellent. Rather, it reflects the fact that the original submission had a foundational weakness that, while you've now developed a fantastic solution for, requires a level of revision that falls outside the scope of a rebuttal.
> > > >
> > > > This is a classic borderline paper for me. The research is strong, the execution is thorough, and the effort you've shown here is exemplary. I genuinely believe that if you integrate these new environments and the corresponding analysis into a revised version of the paper, it would be a clear accept. I truly hope to see this work submitted again to a top venue.
> > > >
> > > > Thank you again for the detailed and engaging discussion. It has been a pleasure reviewing your work.

---

> > > > > ### Author Response · Authors · 2025-08-09
> > > > >
> > > > > Thank you for your responses! We are glad to know that the two additional environments meet your expectations of visual indispensability. Your suggestions have been very helpful in strengthening our work, and it has been a pleasure to discuss with you.
> > > > >
> > > > > For your suggestions on the revision of our paper's logic and structure, we agree that these changes are hard to evaluate without a revised manuscript. Regardless of whether our score is adjusted, we sincerely hope to find a way to keep improving the quality of our work, and we plan to make the following revisions to our manuscript.
> > > > >
> > > > > 1. Introduction: Revise the third paragraph (L38-47) to discuss the motivations for using visual observation in different environments. More specifically: "On the one hand, visual observations are indispensable in some environments like video games with critical and fast-changing spatio cues that are hard to capture by test. On the other hand, many strategic domains such as board games and card games naturally rely on rich visual representation rather than text strings."
> > > > > 2. Analysis of the new results in relation to others: We find these visually indispensable environments are especially challenging for VLMs. For the best-performing VLM gemini-2.5-flash, the average return in these three environments is 16.1, while the average return in the other environments is 29.8. The same results are observed in other VLMs. Also, most non-reasoning VLMs perform worse than random agents in *Pong* and *KAZ*, showing their incompetence in environments with visual indispensability.
> > > > > 3. Overall conclusion: In general, the main findings remain the same: existing VLMs are bad at strategic reasoning and decision-making. Looking deeper, the visually indispensable environments are especially challenging because of the coupled difficulty of visual perception and strategic interactions.
> > > > >
> > > > > We will also revise the other corresponding parts in our manuscript to discuss the new insights provided by the additional environments. We understand that the content in this response is not a full substitute for a complete revised manuscript to better assess the improved paper. We sincerely thank you for the time and effort you have devoted to reviewing our work.

---

### Official Review · Reviewer_5dwK · 2025-07-03

**Rating:** 4
**Confidence:** 3

**Summary:**

This paper introduces VS-Bench, a multimodal benchmark designed to evaluate the strategic reasoning and decision-making capabilities of Vision-Language Models (VLMs) in multi-agent environments.

**Dataset Code Accessibility:**

Yes

**Ethical Considerations:**

No, there are no or only very minor ethics concerns

**Final Justification:**

The authors have answered most of my concerns well, and more evidences have been provided to show the effectiveness of the benchmark.

**Limitations Weaknesses:**

- While VS-Bench covers various interaction types, whether the eight selected games can fully represent all complex strategic reasoning scenarios, and whether the results can generalize broadly to other practical applications, is worth further consideration.
- The paper reveals the shortcomings of current VLMs in these tasks, but there is still room for further exploration regarding the specific reasons for model failures and how to fundamentally improve their strategic capabilities.
- Although the paper does not directly discuss it, the design of any benchmark can unintentionally introduce biases, for example, certain game choices favoring specific models, or evaluation metrics failing to capture all important strategic behaviors.

**Strengths Contributions:**

Existing VLM benchmarks are mainly limited to single-agent or text-only environments, whereas real-world scenarios often involve multiple agents interacting under rich visual and linguistic observations. VS-Bench aims to bridge this gap by evaluating VLM agents' multimodal perception and strategic interaction capabilities in multi-agent settings. VS-Bench comprises eight vision-grounded environments, covering cooperative, competitive, and mixed-motive interactions, designed to assess agents' ability to infer other agents' future moves and optimize long-term objectives. These games include:
- Hanabi and Overcooked, requiring agents to understand teammates' intentions and coordinate actions to achieve common goals.
- Breakthrough, Kuhn Poker, and Atari Pong, requiring agents to model opponents and maintain adversarial exploitation capabilities.
- Coin Dilemma, Monster Hunt, and Battle of the Colors, requiring agents to balance conflicting interests and maintain cooperation while avoiding exploitation.

---

> ### Author Rebuttal · Authors · 2025-07-31
>
> Thank you for your constructive comments and questions! We hope the following response can address your concerns.
>
> ---
>
> > W1: Whether the eight selected games can fully represent all complex strategic reasoning scenarios, and whether the results can generalize broadly to other practical applications, is worth further consideration.
>
> We would like to clarify that our goal with VS-Bench is not to exhaustively cover all possible strategic scenarios. In fact, no benchmark with finite environments can achieve this. Instead, we aim to provide a representative suite of games that span key game-theoretic axes and multi-agent abilities, as done in prior publications like [1] with 6 games, [2] with 7 games, and [3] with 11 games.
>
> To show that our 8 games cover a representative suite of strategic scenarios, we provide a detailed taxonomy of our games, considering
> * Game-theoretic properties: interaction type, full/partial observation, deterministic/probabilistic transition, simultaneous/sequential move.
> * Multi-agent abilities: spatial reasoning, numerical reasoning, opponent modelling, long-term planning, risk assessment, and team collaboration.
>
> In addition, our framework provides a generic and extensible interface, facilitating integration of additional multi-agent games.
>
> | | Interaction | Full Obs. | Deterministic | Simultaneous | Spatial Reasoning | Numerical Reasoning | Opponent Modeling | Long-Term Planning | Risk Assessment | Team Collaboration |
> |:-:|:-:|:-:|:-:|:-:|:-:|:-:|:-:|:-:|:-:|:-:|
> | *Hanabi* | cooperative | ✗ | ✗ | ✗ | low | middle | high | high | middle | high |
> | *Overcooked* | cooperative | ✓ | ✓ | ✓ | high | low | middle | middle | low | high |
> | *Breakthrough* | competitive | ✓ | ✓ | ✗ | high | low | middle | high | high | low |
> | *Poker* | competitive | ✗ | ✗ | ✗ | low | high | middle | low | high | low |
> | *Pong* | competitive | ✓ | ✗ | ✓ | high | low | low | low | middle | low |
> | *Dilemma* | mixed-motive | ✓ | ✗ | ✓ | middle | middle | high | middle | middle | middle |
> | *Hunt* | mixed-motive | ✓ | ✗ | ✓ | middle | middle | high | middle | high | middle |
> | *Battle* | mixed-motive | ✓ | ✗ | ✓ | middle | middle | high | middle | middle | middle |
>
> For the generalization of results to other applications, we argue that our main findings are consistent across 8 environments and are expected to extend to other multi-agent games.
> * Strategic reasoning: in all games, most VLMs outperform random, and most VLMs' accuracy is below 60%, which shows their incompetence in strategic reasoning.
> * Decision-making: in all games except *Poker*, most VLMs' normalized return is below 30%, which shows their poor performance in decision-making.
>
> [1] Wang, Xinyu, Bohan Zhuang, and Qi Wu. "Are Large Vision Language Models Good Game Players?." ICLR 2025.
>
> [2] Chen, Junzhe, et al. "Llmarena: Assessing capabilities of large language models in dynamic multi-agent environments." ACL 2024.
>
> [3] Duan, Jinhao, et al. "Gtbench: Uncovering the strategic reasoning capabilities of llms via game-theoretic evaluations." NeurIPS 2024.
>
>
> > W2: The paper reveals the shortcomings of current VLMs in these tasks, but there is still room for further exploration regarding the specific reasons for model failures and how to fundamentally improve their strategic capabilities.
>
> In Section 4.3 and Appendix G of our manuscript, we provide 6 detailed examples and a qualitative analysis of the failure reasons in different games. To quantitatively evaluate the failure modes, we provide a preliminary breakdown of the failure modes based on the required abilities.
>
> | | Spatial Reasoning | Numerical Reasoning | Opponent Modeling | Long-Term Planning | Risk Assessment | Team Collaboration |
> |:-:|:-:|:-:|:-:|:-:|:-:|:-:|
> | *Hanabi* | 0% | 0% | 40% | 30% | 20% | 10% |
> | *Overcooked* | 70% | 0% | 10% | 15% | 0% | 5% |
> | *Breakthrough* | 30% | 0% | 0% | 30% | 40% | 0% |
> | *Poker* | 0% | 30% | 20% | 20% | 30% | 0% |
> | *Pong* | 40% | 0% | 0% | 30% | 30% | 0% |
> | *Dilemma* | 10% | 0% | 20% | 10% | 10% | 50% |
> | *Hunt* | 10% | 0% | 20% | 0% | 20% | 50% |
> | *Battle* | 10% | 0% | 30% | 10% | 0% | 50% |
>
> On how to fundamentally improve their strategic capabilities, we think there are several different methods.
> 1. Prompt engineering: the easiest but not generalizable way to improve model performance by designing task-specific prompts.
> 2. Workflow design: equip models with external modules and workflows like k-level reasoning, tree search, etc., to improve performance.
> 3. Model fine-tuning: fine-tune models with collected data and utilize reinforcement learning to incentivize strategic ability.
>
> We provide a preliminary result of improving performance by prompting VLMs with cooperative personas (maximize group's total return) in mixed-motive games. Further design and investigation of other improving methods are for future research based on our benchmark.
>
> | | `o4-mini` | `gemini-2.5-flash` | `doubao-1-5-thinking-pro` |
> |:-:|:-:|:-:|:-:|
> | *Dilemma* | -4.6 (21.4) | 10.0 (25.5) | 0.7 (3.2) |
> | *Dilemma* cooperative | 72.8 (11.1) | 53.7 (10.4) | 15.7 (7.1) |
> | *Hunt* | 24.9 (8.2) | 26.2 (5.8) | 17.2 (11.3) |
> | *Hunt* cooperative | 57.1 (22.7) | 15.9 (16.4) | 15.7 (27.7) |
> | *Battle* | 3.5 (5.4) | 32.8 (8.5) | 4.0 (4.8) |
> | *Battle* cooperative | 19.2 (10.5) | 22.2 (9.1) | 6.1 (6.0) |
>
>
> > W3: Although the paper does not directly discuss it, the design of any benchmark can unintentionally introduce biases, for example, certain game choices favoring specific models, or evaluation metrics failing to capture all important strategic behaviors.
>
> We agree with the reviewer that the design of any benchmark can unintentionally introduce biases. In our benchmark, we observe that the results in Table 1 and Table 2 show that the performance of different models is mostly consistent across different games. More specifically,
> * For strategic reasoning: `o4-mini`, `gemini-2.5-flash` and `claude-3-7-sonnet` consistently rank as top 3 in most games. Also, reasoning models are consistently better than chat models and open-source models.
> * For decision-making: `gemini-2.5-flash`, `o4-mini`, `doubao-1-5-thinking-pro`, `claude-3-7-sonnet` consistently rank as top 4 in most games. Chat models and open-source models struggle to outperform random agents in most games.
>
> Though the results show the performance of different models is mostly consistent across our environments, we acknowledge that bias is inevitable in benchmark design and will add a discussion in the Limitations section.
>
> For evaluation metrics, we adopt the common practice of using episode return to evaluate decision-making in multi-agent tasks, as done in prior publications like [1, 2, 3]. We further consider an additional evaluation metric of next-action prediction accuracy to evaluate the strategic reasoning ability of VLM agents. Compared to prior publications, we adopt more comprehensive evaluation metrics, which we believe constitute a solid evaluation and meaningful contribution.
>
> ---
>
> We genuinely value your dedication to reviewing our paper. We have carefully addressed each of your concerns, and we sincerely hope our efforts merit a raise in your rating.

---

> > ### Comment · Reviewer_5dwK · 2025-08-05
> >
> > Thanks for the author's rebuttal, and it has solved most of my concerns. Further the proposed benchmark could help the others to evaluate their methods or models on VLM. I will raise my score.

---

### Official Review · Reviewer_Saza · 2025-07-06

**Rating:** 3
**Confidence:** 4

**Summary:**

The paper describes a multimodal benchmark incorporating eight vision-grounded games across cooperative, competitive, and mixed-motive settings. The benchmark is designed to evaluate Vision-Language Models (VLMs) for strategic reasoning and decision-making in multi-agent environments. It evaluates VLMs through by the following two: offline strategic reasoning (next-action prediction accuracy)
and online decision-making (normalized episode return). The benchmark tests 14 VLMs (including open-source and commercial models) and finds a substantial gap between current model performance and optimal strategies. The best models achieved only 45.8% prediction accuracy and 26.3% normalized return. Failure analyses and ablations reveal challenges in multimodal perception, reasoning under uncertainty, and social dilemma navigation.

**Dataset Code Accessibility:**

Yes

**Dataset Code Comments:**

-All games, prompts, evaluation metrics, and failure analysis are thoroughly documented in the main paper and appendices.

-All experiment protocols, metrics and evaluation procedure are defined in the appendix.

-Models are evaluated bia both APIs and open-source libraries.

**Ethical Considerations:**

No, there are no or only very minor ethics concerns

**Final Justification:**

This paper presents a well-motivated and thoroughly executed benchmark for evaluating vision-language models (VLMs) in multi-agent strategic environments. I appreciate the authors' thoughtful design of the benchmark, their dual evaluation approach (offline prediction and online decision-making), and their extensive experiments across 14 VLMs. The added human-agent comparison, theory-of-mind case study, and detailed game-theoretic breakdown meaningfully strengthen the work.

That said, I am maintaining my borderline reject score due to a key unresolved concern: the paper’s central framing as a visual benchmark. Most of the environments rely on symbolic game logic where the visual modality is not strictly necessary. As the authors’ own ablations show, models often perform better with text-only input, suggesting the benchmark primarily tests symbolic reasoning under perceptual noise—rather than uniquely visual strategic reasoning. While this is a valid evaluation setting, it does not fully align with the benchmark's stated contribution.

I believe this is a valuable and timely effort, and with either a reframed focus on multi-agent reasoning or the integration of truly visual environments (as proposed in the rebuttal), it could become a strong and impactful benchmark. I look forward to seeing its future evolution.

**Limitations Weaknesses:**

- The benchmark considers only two-agent settings.
-The paper does not provide human-agent comparison.
- Strategic reasoning is assessed only via prediction accuracy, which may miss nuance in strategy diversity. Precision/recall per action could be more informative.
-The paper achieves does achieve high performance: less than 50% accuracy and 30% return.

**Strengths Contributions:**

-It claims it is the first benchmark for multimodal multi-agent strategic reasoning, combining vision, language, and interaction.
-The benchmark can measure both theory-of-mind (ToM) capabilities (via next-action prediction) and long-term planning (via normalized return).
-The benchmark is obtained from eight environments (e.g., Hanabi, Overcooked, Kuhn Poker, Monster Hunt), well-chosen from game theory and MARL, representing real-world strategic challenges.
-The validity of the benchmark is evaluated on 14 VLMs, including open-source models (e.g., InternVL3, LLaMA 3.2) and proprietary systems (e.g., GPT-4.1, Claude 3.7, Gemini 2.5).
-The paper includes ablation studies on vision and CoT prompting, behavioral analysis in social dilemmas, and detailed failure case analyses.

---

> ### Author Rebuttal · Authors · 2025-07-31
>
> Thank you for your constructive comments and questions! We hope the following response can address your concerns.
>
> ---
>
> > W1: The benchmark considers only two-agent settings.
>
> Our benchmark is a general framework that can be applied to n-player games with n > 2. To validate this claim, we perform experiments on 2 additional environments: (1) 3-player Hanabi and (2) 3-player Coin Dilemma. We chose several representative models from the 14 VLMs, and the results are as follows.
>
> | | `gemini-2.5-flash` | `o4-mini` | `doubao-1.5-thinking-pro-m` | `gpt-4.1` | `gemini-2.5-flash` w/o thinking | `doubao-1.5-vision-pro` | `Qwen2.5-VL-72B-Instruct` | `Llama-3.2-90B-Vision-Instruct` |
> |:-:|:-:|:-:|:-:|:-:|:-:|:-:|:-:|:-:|
> | 3-player Hanabi | 9.2 (2.6) | 12.2(1.31) | 10.4 (4.0) | 3.8 (1.4) | 2.3 (1.4) | 4.3 (1.4) | 0.4 (0.5) | 1 (1.1) |
> | 3-player Coin Dilemma | -15.3 (13.7) | -24.91(21.22) | 1.1 (10.4) | 5.3 (8.9) | -3.6 (10.9) | -2.1 (5.0) | -4.6 (8.9) | -1.1 (5.1) |
>
> Our work mainly focuses on 2-agent settings because it is the most canonical and widely studied case in multi-agent research, which serves as a foundational step toward more general n-agent evaluation. We believe our unified benchmark, comprehensive evaluation metrics, and extensive experiment results already constitute a solid and sufficient contribution, as evidenced by prior publications like [1] in NeurIPS 2024 and [2] in ICLR 2025, which also only consider 2-agent settings.
>
> [1] Duan, Jinhao, et al. "Gtbench: Uncovering the strategic reasoning capabilities of llms via game-theoretic evaluations."  NeurIPS 2024.
>
> [2] Wang, Xinyu, Bohan Zhuang, and Qi Wu. "Are Large Vision Language Models Good Game Players?." ICLR 2025.
>
> > W2: The paper does not provide human-agent comparison.
>
> To establish a human baseline for better assessing VLMs' performance, we recruited about 25 human participants to perform the decision-making tasks. For a fair comparison, humans were given exactly the same observations as the models, and we collected the same number of trajectories in each environment as the VLMs. The experiment results are shown below, and we also include the results of the best-performing VLM `gemini-2.5-flash` for comparison.
>
> | | Overall | *Hanabi* | *Overcooked* | *Board* | *Poker* | *Pong* | *Dilemma* | *Hunt* | *Battle* |
> |:-:|:-:|:-:|:-:|:-:|:-:|:-:|:-:|:-:|:-:|
> | humans | 71.1 | 72.9 (27.0) | 87.9 (21.5) | 100.0 (0.0) | 46.5 (33.2) | 76.0 (27.6) | 38.1 (54.9) | 66.0 (32.0) | 81.7 (34.3) |
> | `gemini-2.5-flash` | 26.3 | 27.1 (36.0) | 8.5 (5.4) | 20.0 (51.5) | 84.1 (19.9) | 1.6 (1.9) | 10.0 (25.5) | 26.2 (5.8) | 32.8 (8.5) |
>
> As shown in the results, humans achieve a **71.1%** overall normalized return, much stronger than the 26.3% return obtained by `gemini-2.5-flash`. This further validates our findings that existing VLMs struggle in multi-agent tasks.
>
> > W3: Strategic reasoning is assessed only via prediction accuracy, which may miss nuance in strategy diversity. Precision/recall per action could be more informative.
>
> We would like to argue that accuracy is the de facto standard metric for reasoning tasks. Mainstream benchmarks like MMLU [3], GPQA [4], GSM8K [5], etc., and multi-agent benchmarks like LVLM-Playground [2] all evaluate models by accuracy. Accordingly, we regard accuracy as a suitable and informative metric for our strategic reasoning tasks.
>
> [3] Hendrycks, Dan, et al. "Measuring massive multitask language understanding." arXiv preprint arXiv:2009.03300 (2020).
>
> [4] Rein, David, et al. "Gpqa: A graduate-level google-proof q&a benchmark." First Conference on Language Modeling. 2024.
>
> [5] Cobbe, Karl, et al. "Training verifiers to solve math word problems." arXiv preprint arXiv:2110.14168 (2021).
>
> > W4: The paper does not achieve high performance: less than 50% accuracy and 30% return.
>
> We would like to clarify that our work does not propose a method to achieve high performance in multi-agent environments. Instead, we provide a benchmark to evaluate existing VLMs and show their limitation in multi-agent tasks. Therefore, the unsatisfactory performance of existing VLMs should not be viewed as a weakness of our work. Rather, it exposes the gap in existing VLMs' ability in multi-agent tasks and demonstrates the value of our benchmark.
>
> ---
>
> We genuinely value your dedication to reviewing our paper. We have carefully addressed each of your concerns, and we sincerely hope our efforts merit a raise in your rating.

---

> ### Comment · Reviewer_Saza · 2025-08-06
> **Final justification**
>
> This paper presents a well-motivated and thoroughly executed benchmark for evaluating vision-language models (VLMs) in multi-agent strategic environments. I appreciate the authors' thoughtful design of the benchmark, their dual evaluation approach (offline prediction and online decision-making), and their extensive experiments across 14 VLMs. The added human-agent comparison, theory-of-mind case study, and detailed game-theoretic breakdown meaningfully strengthen the work.
>
> That said, I am maintaining my borderline reject score due to a key unresolved concern: the paper’s central framing as a visual benchmark. Most of the environments rely on symbolic game logic where the visual modality is not strictly necessary. As the authors’ own ablations show, models often perform better with text-only input, suggesting the benchmark primarily tests symbolic reasoning under perceptual noise—rather than uniquely visual strategic reasoning. While this is a valid evaluation setting, it does not fully align with the benchmark's stated contribution.
>
> I believe this is a valuable and timely effort, and with either a reframed focus on multi-agent reasoning or the integration of truly visual environments (as proposed in the rebuttal), it could become a strong and impactful benchmark. I look forward to seeing its future evolution.

---

### Official Review · Reviewer_AWRM · 2025-07-13

**Rating:** 4
**Confidence:** 4

**Summary:**

This paper proposed a benchmark called VS-Bench, which evaluates Vision-Language Models in strategic, multi-agent settings. Across eight vision-based cooperative, competitive and mixed-motive games, it measures both strategic reasoningand long-term decision-making. Experiments on fourteen state-of-the-art VLMs are conducted to show a large gap to optimal performance.

**Dataset Code Accessibility:**

Yes

**Ethical Considerations:**

No, there are no or only very minor ethics concerns

**Final Justification:**

Thanks for the author's rebuttal, and it solved part of my concern. Similar to the opinion of Reviewer q2zX, I believe this benchmark is more like a combination of perception and decision-making rather than pure visual strategic reasoning. Therefore, I think this is borderline work, and I will maintain my original rating.

**Limitations Weaknesses:**

1. The details of multi-agent setting are unclear.
2. There should be some evaluations of existing state-of-the-art agent models to show the upper bound of the benchmark.
3. The domains in the benchmark are very limited, mostly pixel-based games. The experiments also show that performance based purely on text is sometimes even better, which indicates that existing evaluations may not effectively assess the agent's capabilities and instead are limited to visual abilities, especially the ability to process visual information in certain specific domains.
4. Many tasks are based on the existing datasets, which limits the contribution of this benchmark.

**Strengths Contributions:**

1. The details of problem setting are clearly presented.
2. The chart clearly and explicitly conveys the information.

---

> ### Author Rebuttal · Authors · 2025-07-31
>
> Thank you for your appreciation of our work and constructive feedback. We hope the following response can address your concerns.
>
> ---
>
> > W1: The details of multi-agent setting are unclear.
>
> We consider two kinds of multi-agent settings, and their details are as follows:
> 1. Simultaneous move: in each step, all agents receive their own observations and choose their actions at the same time, without knowing the actions of other agents. This setting includes *Overcooked*, *Pong*, *Dilemma*, *Hunt*, and *Battle*.
> 2. Sequential move: agents act in a specific order, with later agents able to (partially) observe the earlier agents' actions. This setting includes *Hanabi*, *Breakthrough*, and *Poker*.
>
> More formally, the multi-agent games are formulated as Partially Observable Markov Games (POMG), which are described in detail in Section 2.1 of our manuscript. We are more than happy to provide more explanations if you have further questions about the details of our multi-agent settings.
>
> > W2: There should be some evaluations of existing SOTA agent models.
>
> By the time of our submission, there were few available agent models. In our manuscript, we evaluated `o4-mini`, which can be considered a model with agentic ability, as OpenAI described in their [release blog](https://openai.com/index/introducing-o3-and-o4-mini/): 'for the first time, our reasoning models can agentically use and combine every tool within ChatGPT'
>
> We additionally evaluated 3 models: `o3` and `gemini-2.5-pro` are the SOTA reasoning models with agent ability, and `UI-TARS-1.5` is a SOTA agent model in computer use and games. The main findings remain the same: existing models achieve < reasoning accuracy and < 36% return in decision-making, which are far from optimal in multi-agent environments.
>
> | | Overall | *Hanabi* | *Overcooked* | *Board* | *Poker* | *Pong* | *Dilemma* | *Hunt* | *Battle* |
> |:-:|:-:|:-:|:-:|:-:|:-:|:-:|:-:|:-:|--------------|
> | `o3` | 35.3 | 55.8 (21.0) | 15.6 (3.0) | 80.0 (40.0) | 81.8 (18.2) | 8.6 (16.1) | -0.4 (17.4) | 24.0 (3.2) | 16.7 (5.3) |
> | `gemini-2.5-pro` | 23.1 | 32.9 (35.5) | 17.1 (5.6) | 55.0 (49.5) | 27.3 (54.5) | 6.5 (11.4) | -9.6 (19.7) | 21.5 (6.0) | 33.8 (6.5) |
> | `UI-TARS-1.5` | 14.3 | 0.0 (0.0) | 2.0 (3.4) | 5.0 (15.0) | 36.4 (27.3) | -0.9 (0.3) | 33.1 (14.3) | 24.9 (3.3) | 13.6 (2.9) |
>
> Some other models are not evaluated because:
> * OpenAI and Claude `computer-use`: designed for computer use and sends actions like click and type, which are not compatible with our environments.
> * ChatGPT Agent and Kimi k2: released after our submission (2025.5.15).
>
> > W3: Existing evaluations may not effectively assess the agent's capabilities and instead are limited to visual abilities, especially the ability to process visual information in certain specific domains.
>
> We would like to argue that the fundamental reason for existing models' poor performance in our evaluations is not their limited perception capabilities, but rather their insufficient reasoning and decision-making abilities. To show that we perform 2 additional experiments.
>
> (1) Perception: Existing models are good at visual perceptions, achieving **70-80%** overall accuracy. However, they still struggle in reasoning (<46%) and decision-making (<27%).
>
> We evaluate the perception ability of VLMs by asking them to output a dict or a matrix representation of all information in the image (board state, card number, etc) and report their accuracy. *Pong* is not evaluated because its states are pixel-based and hard to represent by text.
>
> | | Overall | *Hanabi* | *Overcooked* | *Breakthrough* | *Poker* | *Dilemma* | *Hunt* | *Battle* |
> |:-:|:-:|:-:|:-:|:-:|:-:|:-:|:-:|:-:|
> | `gemini-2.5-flash` | 80.3 | 79.9 | 54.5 | 98.5 | 100.0 | 76.4 | 73.0 | 79.9
> | `o4-mini` | 85.3 | 79.7 | 69.8 | 97.2 | 99.1 | 85.5 | 80.2 | 85.4 |
> | `doubao-1-5-thinking-pro` | 73.7 | 42.2 | 52.4 | 91.0 | 98.6 | 76.7 | 74.9 | 80.0 |
> | `claude-3-7-sonnet` | 79.8 | 73.0 | 62.8 | 75.2 | 99.5 | 82.6 | 79.7 | 85.7 |
> | `qvq-max` | 76.6 | 75.1 | 63.6 | 83.3 | 95.2 | 69.5 | 72.0 | 77.7 |
> | `gemini-2.5` w/o thinking | 82.9 | 79.9 | 38.8 | 88.2 | 97.2 | 92.5 | 93.1 | 90.8 |
> | `gpt-4.1` | 76.5 | 72.1 | 62.0 | 67.0 | 100.0 | 76.7 | 76.8 | 81.2 |
> | `qwen-vl-max` | 80.0 | 76.1 | 68.2 | 81.2 | 99.2 | 78.4 | 76.4 | 80.6 |
> | `claude-3-7` w/o thinking | 79.9 | 75.9 | 59.7 | 79.0 | 99.6 | 81.8 | 80.4 | 82.8 |
> | `grok-2-vision` | 70.3 | 75.2 | 46.8 | 80.3 | 59.5 | 76.4 | 73.3 | 81.0 |
> | `doubao-1-5-vision-pro` | 77.0 | 80.0 | 33.1 | 89.3 | 100.0 | 78.0 | 77.6 | 81.2 |
> | `Qwen2.5-VL-72B-Ins.` | 80.7 | 76.0 | 72.9 | 75.1 | 100.0 | 79.8 | 79.0 | 82.4 |
> | `InternVL3-78B` | 74.5 | 74.6 | 43.6 | 64.3 | 99.2 | 81.1 | 76.7 | 81.8 |
> | `Llama-3.2-90B-Vision-Ins` | 62.5 | 30.7 | 58.6 | 59.7 | 81.6 | 68.5 | 66.0 | 72.4 |
>
> (2) Text-only decision-making: without visual perception, existing models still struggle in decision-making, achieving **<36%** normalized return.
>
> | | Overall | *Hanabi* | *Overcooked* | *Board* | *Poker* | *Dilemma* | *Hunt* | *Battle* |
> |:-:|:-:|:-:|:-:|:-:|:-:|:-:|:-:|:-:|
> | `gemini-2.5-flash` | 22.5 | 40.8 (21.9) | 61.3 (19.7) | 30.0 (42.2) | 28.2 (60.9) | -26.3 (18.1) | 3.4 (12.8) | 20.2 (5.7) |
> | `o4-mini` | 27.1 | 37.1 (26.1) | 58.8 (23.3) | 30.0 (47.0) | 79.1 (33.6) | -42.7 (22.4) | 2.8 (8.4) | 24.8 (13.7) |
> | `doubao-1-5-thinking-pro` | 29.7 | 37.5 (32.9) | 80.9 (30.6) | 15.0 (37.0) | 62.7 (41.8) | -22.8 (19.6) | 13.5 (7.3) | 21.2 (7.2) |
> | `claude-3-7-sonnet` | 35.9 | 33.8 (35.8) | 81.9 (26.8) | 45.0 (50.0) | 76.4 (42.7) | -37.4 (18.4) | 11.8 (15.7) | 39.7 (10.4) |
> | `qvq-max` | 3.2 | 0.0 (0.0) | 23.1 (13.1) | 5.0 (15.8) | -22.7 (55.5) | -1.8 (12.2) | 9.4 (8.2) | 9.6 (8.1) |
> | `gemini-2.5` w/o thinking | 4.8 | 0.0 (0.0) | 1.0 (2.3) | 0.0 (0.0) | 2.7 (10.9) | 23.8 (13.3) | 4.1 (8.6) | 2.0 (3.4) |
> | `gpt-4.1` | 11.7 | 0.0 (0.0) | 0.0 (1.5) | 0.0 (0.0) | 1.8 (38.2) | 40.2 (12.0) | 18.4 (15.6) | 21.7 (12.4) |
> | `qwen-vl-max` | 3.2 | 0.0 (0.0) | 1.0 (2.3) | 5.0 (15.8) | 1.8 (60.9) | 3.6 (5.0) | 10.7 (14.7) | 0.0 (1.5) |
> | `claude-3-7` w/o thinking | 9.6 | 0.0 (0.0) | 4.5 (5.5) | 5.0 (15.8) | 0.9 (23.6) | 35.6 (18.1) | 3.5 (6.9) | 17.7 (10.2) |
> | `grok-2-vision` | 1.8 | 0.0 (0.0) | 0.0 (1.5) | 0.0 (0.0) | 10.9 (55.5) | -1.1 (1.8) | 0.9 (8.1) | 1.5 (2.5) |
> | `doubao-1-5-vision-pro` | 1.3 | 0.0 (0.0) | -0.5 (0.0) | 10.0 (21.0) | -13.6 (24.6) | 5.3 (13.5) | 7.8 (8.8) | 0.0 (1.5) |
> | `Qwen2.5-VL-72B-Ins.` | 2.9 | 6.2 (6.6) | -0.5 (0.0) | 0.0 (0.0) | 2.7 (36.4) | 2.1 (3.6) | 9.1 (14.6) | 0.5 (2.0) |
> | `InternVL3-78B` | 0.8 | 0.0 (0.0) | 3.0 (3.9) | 0.0 (0.0) | -7.3 (45.5) | 3.2 (3.0) | 5.6 (4.6) | 1.2 (2.0) |
> | `Llama-3.2-90B-Vision-Ins` | 22.5 | 40.8 (21.9) | 61.3 (19.7) | 30.0 (42.2) | 28.2 (60.9) | -26.3 (18.1) | 3.4 (12.8) | 20.2 (5.7) |
>
> The above 2 additional experiment results show that existing VLMs are good at perception, but poor at reasoning and decision-making in our environments. Therefore, our evaluations effectively assess agents' reasoning and decision-making ability.
>
> > W4: Many tasks are based on the existing datasets, which limits the contribution of this benchmark.
>
> One of our contributions is the multimodal benchmark with 8 environments for evaluating VLMs. Although some of the environments have existing implementations, they cannot be directly used to evaluate VLM agents. We made the following efforts to build a benchmark for VLM agents:
> 1. Unified framework: we developed a unified framework to evaluate VLM agents in these environments.
> 2. Visualization and prompt: we implemented the visualization in 6 out of 8 environments and the prompt design for all environments.
> 3. Reasoning dataset: we provide a new dataset for strategic reasoning, and the data in 7 out of 8 environments are collected by ourselves.
>
> Besides the first contribution, our work also has the following two contributions
> 1. Evaluation metrics: we consider both offline evaluation of strategic reasoning and online evaluation of decision-making to provide a comprehensive assessment.
> 2. Extensive experiments: we conduct extensive experiments of 14 VLMs and provide in-depth analysis.
>
> Therefore, our work is well beyond merely reusing existing environments and makes a sufficient contribution to the field with a unified framework, a new dataset, comprehensive metrics, and extensive experiments.
>
> ---
>
> We would like to express our appreciation for your feedback and hope our answers have addressed your concerns. We would be very grateful if you could consider raising the rating of our work based on our responses.

---

> > ### Comment · Reviewer_AWRM · 2025-08-04
> > **Feedback after rebuttal**
> >
> > Thanks for the author's rebuttal, and it solved part of my concern. Similar to the opinion of Reviewer q2zX, I believe this benchmark is more like a combination of perception and decision-making rather than pure visual strategic reasoning. Therefore, I think this is borderline work, and I will maintain my original rating.

---

### Decision · Program_Chairs · 2025-09-18

**Decision:**

Reject

**Comment:**

The paper proposed a multimodal benchmark that consists of 8 vision-grounded environments and aims to evaluate strategic reasoning and decision-making of VLMs in multi-agent settings. The paper received four reviews with mixed ratings: 2x borderline accept and 2x borderline reject. In general, the reviewers appreciated the multi-agent problem setting and extensiveness of the evaluation. They also highlighted that this is a “timely and well-executed work”. The main negative arguement shared by both reviewers Saza and q2zX is that framing the work as a visual benchmark may not be convincing because text-only models sometimes perform better (sec 4.1), suggesting that the nature of the benchmark is more of a combination of visual understanding and strategic reasoning. After carefully evaluating the paper and discussing with the SAC, the AC finds that reviewers Saza and q2zX's concern is valid. The AC also agrees with the comment that a human baseline is necessary to be added to the benchmark so it is easier to interpret the results. Overall, the work has strengths but the weaknesses are notable. The AC recommends rejection.